# Heterotypic interactions can drive selective co-condensation of prion-like low-complexity domains of FET proteins and mammalian SWI/SNF complex

Richoo B. Davis [1], Anushka Supakar[2], Aishwarya Kanchi Ranganath[3], Mahdi Muhammad Moosa [1] & Priya R. Banerjee [1,2,3] ✉

Prion-like domains (PLDs) are low-complexity protein sequences enriched within nucleic acid-binding proteins including those involved in transcription and RNA processing. PLDs of FUS and EWSR1 play key roles in recruiting chromatin remodeler mammalian SWI/SNF (mSWI/SNF) complex to onco-genic FET fusion protein condensates. Here, we show that disordered low-complexity domains of multiple SWI/SNF subunits are prion-like with a strong propensity to undergo intracellular phase separation. These PLDs engage in sequence-specific heterotypic interactions with the PLD of FUS in the dilute phase at sub-saturation conditions, leading to the formation of PLD co-condensates. In the dense phase, homotypic and heterotypic PLD interactions are highly cooperative, resulting in the co-mixing of individual PLD phases and forming spatially homogeneous condensates. Heterotypic PLD-mediated positive cooperativity in protein-protein interaction networks is likely to play key roles in the co-phase separation of mSWI/SNF complex with transcription factors containing homologous low-complexity domains.

Biomolecular condensates such as stress granules and transcription factories are membrane-less subcellular bodies that form and are regulated via the phase separation of multivalent proteins and nucleic acids[1–3]. The physiological functions of biomolecular condensates range from signaling hubs under normal conditions to storage depots in response to cellular stress[3]. Sequence analyses of the proteins enriched in intracellular condensates both in the nucleus and cyto-plasm have previously revealed an abundance of proteins containing long stretches of intrinsically disordered prion-like domains[4,5]. Prion-like domains (PLDs) are typically characterized by their low complexity sequence features with an overrepresentation of aromatic (Y/F) and polar amino acids (G/S/Q/N) and depletion of charged residues[6,7]. Proteins with PLDs have been identified in all life forms[8–12]. Prion pro-teins were initially discovered as proteinaceous infectious agents in

bovine spongiform encephalopathy and other neurodegenerative diseases[4,13–15], but are increasingly recognized with key functional roles in driving phase separation of RNA-binding proteins in the cell, and in the formation of functional amyloids[16,17].

What roles do PLDs play in the context of protein phase separa-tion? Multivalent cohesive interactions between PLD chains as well as the chain-solvent interactions, which are encoded by the PLD primary sequence composition and patterning[18,19], have been recognized to be a key feature driving the phase separation of isolated PLD chains and PLD-containing proteins[20–25]. Previous studies using hnRNP A1 PLD have demonstrated that the distributed aromatic (Y/F) amino acids act as "stickers" that mediate PLD-PLD interactions[18,26], whereas the polar amino acids (S/G/Q) can be described as "spacers", which regulate chain solvation and cooperativity of sticker-sticker interactions. The

[1]Department of Physics, University at Buffalo, Buffalo, NY 14260, USA. [2]Department of Biological Sciences, University at Buffalo, Buffalo, NY 14260, USA. [3]Department of Chemical and Biological Engineering, University at Buffalo, Buffalo, NY 14260, USA. ✉e-mail: prbanerj@buffalo.edu

importance of tyrosine residues in driving PLD phase separation has been further demonstrated experimentally for FUS[26] and EWSR1[27], and computationally for a large number of PLD sequence variants[28]. In addition to aromatic residues, charged residues such as arginine (R) and polar amino acids such as glutamine (Q) also play smaller but important roles in PLD phase separation[28]. Together, the sticker and spacer residues regulate PLD phase separation in a context-dependent manner[19]. In a broader context, however, the sequence grammar encoding LCD phase separation can be more complex and additional factors beyond aromaticity, such as the net charge and hydrophobicity of LCDs are likely to play equally dominant roles in driving their phase separation[29,30]. When part of a multi-domain protein, π-π and cation-π interactions mediated by the aromatic and arginine residues in PLDs have been shown to drive phase separation of many full-length RNA and DNA binding proteins including FUS, EWSR1, TAF15, hnRNP A1, and EBF1[21,31–34]. Further, debilitating point mutations in PLDs have been reported to promote the pathological transformation of protein condensates from a liquid-like state to solid aggregates[20,35,36]. Thus, PLDs play important roles in the context of functional protein phase separation as well as disease processes associated with the formation of aberrant biomolecular condensates.

Many intracellular biomolecular condensates, such as stress granules and transcriptional hubs, are known to contain a multitude of proteins with [4,5,37]. Despite being broadly classified as prion-like based on the frequencies of certain amino acids in a protein sequence, as noted above[6,7], individual PLD chains typically feature distinct sequence composition, amino acid patterning, and chain length[4,22,38]. Do PLDs from distinct yet functionally related proteins interact with one another and undergo co-phase separation? Previous studies have reported that the PLDs in transcription factors, including the FET family of fusion oncoproteins, not only drive their phase separation but also facilitate the recruitment of essential coactivators, such as the catalytic subunit of the mammalian SWI/SNF (mSWI/SNF) complex, BRG1, in transcriptional condensates[32,36,39–41]. Interestingly, BRG1 contains an N-terminal LCD that is prion-like, which can engage with the PLDs of FET fusion proteins via heterotypic interactions[32]. Although homotypic phase separation of some PLDs, such as FUS and hnRNP A1, are well characterized[18,19,31], little is known about how heterotypic interactions regulate the co-phase separation of PLD mixtures[5] and how the sequence features of respective PLD chains contribute to this process. In general, in multi-component mixtures of multivalent LCDs, homotypic and heterotypic interactions between LCD chains can either positively cooperate, negatively cooperate, or form coexisting phases, resulting in a diverse phase behavior and dense phase co-partitioning. The interplay between the specificity and strengths of homotypic and heterotypic interactions is expected to dictate the co-condensation versus discrete condensate formation in an LCD sequence-specific manner[42]. Further, PLD-containing proteins such as FUS have recently been reported to form a heterogeneous pool of homo-oligomeric complexes below their saturation concentration for phase separation[43], which are thought to represent distinct functional states of the protein than the condensates that form at higher concentrations[44]. However, a key unanswered question is whether heterotypic PLD interactions occur at sub-saturation conditions, which may provide a mechanism for LCD-mediated functional protein networking, such as interactions between FET oncofusions and mSWI/SNF complex, in the absence of phase separation.

Motivated by these open questions, here we systematically investigate the phase behavior of PLD mixtures encompassing FUS[PLD] and the PLDs from the chromatin remodeler mSWI/SNF complex that aberrantly interact with FUS fusion oncoproteins in transcriptional reprogramming[45]. Our study incorporates PLDs from four mSWI/SNF complex subunits: ARID1A, ARID1B, SS18, and BRG1, which are key components for spatiotemporal transcriptional regulation and chromatin remodeling[46–48]. Employing in vitro experiments in conjunction

with mammalian cell culture models, we show that there exists a broad range of saturation concentrations ($C_{sat}$) of PLD chains in vitro that directly correlate with their ability to form phase-separated condensates in live cells. We find that, except BRG1, mSWI/SNF subunit PLDs undergo phase separation with $C_{sat}$ values substantially lower than the known PLDs of RNA-binding proteins such as TAF15, EWSR1, and FUS[18,31], suggesting a greater degree of homotypic interactions. Similar to FET PLDs, the phase separation propensity of mSWI/SNF PLDs is primarily dependent on aromatic residues, specifically tyrosine residues, and to a smaller extent on polar amino acids such as glutamine. Despite strong homotypic interactions, mSWI/SNF PLDs engage in heterotypic interactions with FUS[PLD], resulting in co-partitioning in the dense phase with partition coefficients that show a positive correlation with the number of aromatic residues. In mixtures of PLD condensates, individual PLD phases undergo complete mixing, and together, they form spatially homogeneous PLD co-condensates. These findings indicate that homotypic and heterotypic PLD interactions act cooperatively in the dense phase despite substantially different saturation concentrations of individual PLD chains, which we posit to be a direct manifestation of similarity in PLD sequence grammars. Importantly, heterotypic PLD-PLD interactions between FUS and mSWI/SNF subunits are detectable at sub-saturation concentrations in vitro and in live cells, indicating strong affinities between these low-complexity domains in the absence of phase separation. The observed specificity in interactions among PLDs is further highlighted by a lack of interactions between these PLDs with a functionally distinct non-prion-like LCD. We conjecture that PLD-mediated selective co-condensation of multiple subunits of the mSWI/SNF chromatin remodeling complex with FET fusion proteins may constitute an important step in establishing transcriptionally relevant protein interaction networks.

## Results

### Prion-like domains of mSWI/SNF subunits form dynamic phase-separated condensates in live cells

mSWI/SNF chromatin remodeler complex is enriched in subunits that have large disordered low complexity regions with unknown functions[49]. Many of these disordered regions have prion-like sequences (Fig. S1)[32]. Since PLDs of RNA and DNA binding proteins can drive phase separation and contribute to the formation of biomolecular condensates in cells[4,21,31], we investigated whether mSWI/SNF subunit PLDs are phase separation competent. We selected the top four PLDs in the complex based on their length, functional and disease relevance, which correspond to the following subunits - BRG1 [catalytic subunit], ARID1A, and ARID1B [among most mutated proteins in cancer[48]], and SS18 [relevant to fusion oncoprotein SS18-SSX[50]] (Figs. 1a, b; S1). We noted that although the prion prediction algorithm PLAAC[6] categorizes these low complexity domains as prion-like, these PLDs have varying sequence composition and their lengths are significantly higher than the PLDs from RNA binding proteins (Fig. 1b; Tables S1–3). To determine if they were phase separation competent, we titrated concentrations of recombinant PLDs in vitro (buffer: 125 mM NaCl, 25 mM Tris.HCl pH 7.5) and observed that apart from BRG1[PLD], all other PLDs form spherical condensates in a concentration-dependent manner (Figs. 1c, d; S2a). Further, ARID1B[PLD] condensates showed cluster-like morphologies upon phase separation, suggesting a percolation-type network formation[51] (Fig. 1c). Based on the optical microscopy data, we quantified the saturation concentrations ($C_{sat}$) for the PLDs as ≤2.5 μM for ARID1A[PLD] and SS18[PLD], and ≤ 5.0 μM for ARID1B[PLD] (Figs. 1d; S2a). Under similar experimental conditions, FUS[PLD] undergoes phase separation with a $C_{sat}$ of ≤ 200 μM (Fig. S2a)[52], which is almost two orders of magnitude higher than ARID1A[PLD] and SS18[PLD]. Although BRG1[PLD] did not phase separate under these conditions (Fig. S2a) it can be induced to form spherical condensates in the presence of a macromolecular crowder (20% Ficoll PM70; Fig. S2b)[32].

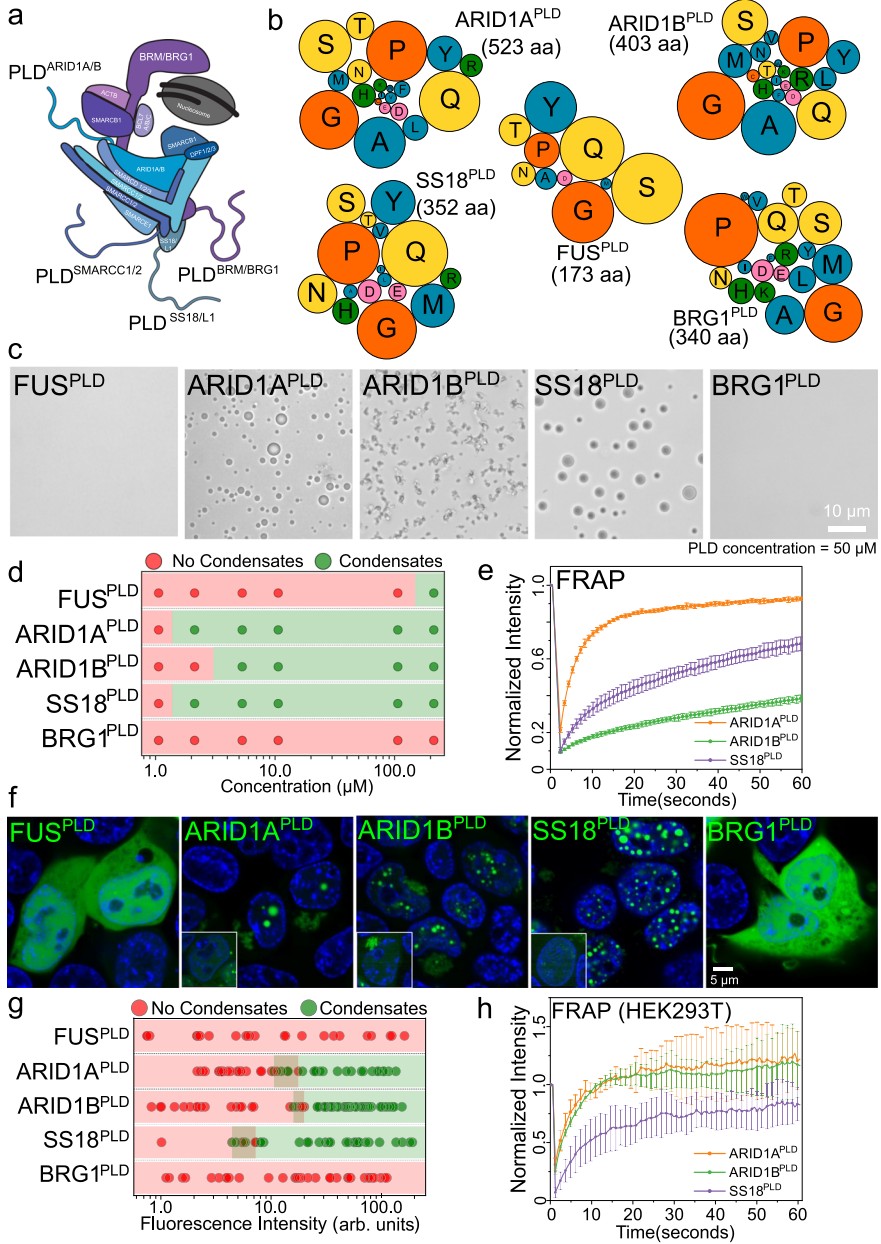

**Fig. 1 | Prion-like domains of mSWI/SNF complex subunits form phase-separated condensates in vitro and in live cells. a** A schematic of the mSWI/SNF complex bound to the nucleosome. This is adapted from Varga et al., BST, 2021[90]. The four largest prion-like domains (PLDs) in the mSWI/SNF complex are displayed as squiggly lines. **b** A bubble chart representation of the sequence composition of the five PLDs (FUS, ARID1A, ARID1B, SS18, and BRG1) used in this study. The lengths of the PLDs are displayed as amino acid count "aa". The color codes for amino acids are provided in Table S6. **c** Differential interference contrast (DIC) microscopy images of purified FUS$^{PLD}$ and mSWI/SNF PLDs at 50 µM protein concentration. **d** Concentration titrations of various PLDs are displayed as state diagrams with green circles denoting the two-phase regime and red circles denoting the single-phase regime (see DIC images in Fig. S2). **e** Fluorescence recovery after photobleaching (FRAP) curves for condensates formed by ARID1A$^{PLD}$, ARID1B$^{PLD}$, and SS18$^{PLD}$ at 50 µM concentration. The FRAP curves show the average intensity and standard deviation of the intensity profiles as a function of time ($n = 4$ condensates

for ARID1A$^{PLD}$ and ARID1B$^{PLD}$, $n = 5$ condensates for SS18$^{PLD}$). **f** Fluorescence microscopy images of HEK293T cells expressing GFP-tagged PLDs (FUS$^{PLD}$, ARID1A$^{PLD}$, ARID1B$^{PLD}$, SS18$^{PLD}$ and BRG1$^{PLD}$), as indicated. Hoechst was used to stain the cell nucleus, which is shown in blue. The insets show images of cells expressing GFP-tagged proteins below their respective saturation concentrations. **g** The relative phase separation capacities were quantified from the fluorescence intensity of intracellularly expressed GFP-tagged proteins (a proxy for protein concentration) and presented as a state diagram. Green circles indicate the presence of nuclear condensates and red circles represent diffused expression patterns. The shaded rectangles represent the transition concentrations (FUS$^{PLD}$ $n = 25$ cells, ARID1A$^{PLD}$ $n = 43$ cells, ARID1B$^{PLD}$ $n = 63$ cells, SS18$^{PLD}$ $n = 33$ cells, and BRG1$^{PLD}$ $n = 34$ cells from two biological replicates). **h** FRAP curves for condensates formed by GFP-tagged PLDs in HEK293T cells. The average intensity and standard deviation of the intensity profiles are shown over time ($n = 3$ cells). The scale bar is 5 µm for all images.

These data suggest that except for BRG1, mSWI/SNF subunit PLDs are highly phase separation competent. We next probed whether these condensates are dynamic using fluorescence recovery after photobleaching (FRAP) experiments. FRAP recovery traces indicate that all PLD condensates have liquid-like properties with varying diffusivity

dynamics. (Fig. 1e). Based on the FRAP traces, we find that ARID1A$^{PLD}$ forms the most dynamic condensates with more than 80% recovery, SS18$^{PLD}$ is intermediate with ~60% recovery, and ARID1B$^{PLD}$ is the least dynamic with less than 40% recovery within the same observational timeframe. The reduced dynamicity of ARID1B$^{PLD}$ condensates is

consistent with the percolation-driven network formation observed for these condensates (Fig. 1c).

Although PLDs have emerged as a driver of many ribonucleo-protein phase separation under physiological and pathological conditions, expression of these domains alone typically does not lead to the formation of condensates in live cells[20,32,42,53]. This is consistent with their known $C_{sat}$ values in vitro, which range from 100 to 200 μM and are typically much higher than their intracellular concentrations[18,31,52,54]. Since mSWI/SNF PLDs show low micromolar $C_{sat}$ values in vitro, we posited that they may form condensates in live cells at relatively low expression levels compared to FUS[PLD]. To test this idea, we transiently transfected HEK293T cells with GFP-PLD plasmids. Upon expression, ARID1A[PLD], ARID1B[PLD], and SS18[PLD] readily formed spherical nuclear foci, whereas BRG1[PLD] and FUS[PLD] remained diffused at all expression levels (Fig. 1f). To estimate the relative $C_{sat}$ of mSWI/SNF PLDs within the nucleus, we used GFP fluorescence intensity as a proxy for concentration and leveraged the stochastic nature of intracellular PLD expression that spanned over two orders of magnitude. We observed that SS18[PLD] has the lowest $C_{sat}$ followed by ARID1A[PLD] and ARID1B[PLD] (Figs. 1g, S3). This rank order of cellular saturation concentrations is similar to their in vitro phase behavior (Fig. 1d). FRAP experiments revealed that the nuclear condensates of ARID1A[PLD], ARID1B[PLD], and SS18[PLD] are dynamic (Fig. 1h). Interestingly, the morphology of the PLD condensates varied with their subcellular localization. Spherical condensates formed within the nucleus and irregular, yet dynamic, assemblies were observed in the cytoplasm (Fig. S3c, d). Such differences could arise from the distinct intracellular microenvironment of the cytoplasm and the nucleus, such as the viscoelasticity of chromatin fibers, altered post-translational modifications, and high abundance of RNAs in the nucleus, which can markedly influence the coarsening behavior and biophysical properties of condensates[55–59].

## Tyrosine residues play a dominant role in mSWI/SNF subunit PLD phase separation

The phase separation capacity of PLDs from RNA-binding proteins has been attributed to multivalent interactions predominantly mediated by the distributed aromatic and arginine residues[18,28]. While SS18[PLD] has a lower fraction of aromatic and arginine residues (0.11) than FUS[PLD] (0.14; Tables S1–3), it possesses a greater number of aromatic and arginine residues (39) than FUS[PLD] (24). To test if increasing the number of aromatic residues can improve the phase separation driving force of FUS[PLD] without changing the overall sequence composition, we created a dimer of FUS[PLD], termed FUS[2XPLD] (Fig. 2a), which possesses a total of 48 aromatic residues at a fixed fraction of 0.14. In contrast to the FUS[PLD], which remained diffused at all expression levels, we observed that FUS[2XPLD] formed phase-separated condensates in the cell nucleus at a relatively low expression level (Fig. 2b) similar to the three mSWI/SNF subunit PLDs (Fig. 1f). The estimated intracellular saturation concentration of FUS[2XPLD] was observed to be similar to that of ARID1A[PLD], ARID1B[PLD], and SS18[PLD] (Figs. 1g, 2b; Figs. S3, S4a). Analogous to mSWI/SNF subunit PLD condensates, FRAP experiments revealed that FUS[2XPLD] condensates have a high degree of dynamic behavior (Fig. S4b). To test whether the stronger driving force for phase separation of FUS[2XPLD] primarily stems from the greater number of tyrosine residues and not simply from its increased length, we further created a variant of FUS[2XPLD], termed FUS[2XPLD halfYtoS], where we replaced the tyrosine residues to serine in the second half of the FUS[2XPLD] (Table S4). This sequence variation led to a complete loss of phase separation of FUS[2XPLD] in living cells even at 10-fold higher intracellular concentrations (Fig. 2a, b; Fig. S3), implying that the number of tyrosine residues is a key determinant of phase separation in this PLD. Next, to test if tyrosine residues are also important for the phase separation of mSWI/SNF subunit PLDs, we first created an ARID1A[PLD] variant where we mutated all 29 tyrosine residues to serine (29Y-to-S), termed

ARID1A[PLD YtoS]. We observed that 29Y-to-S substitution abolished the ARID1A[PLD] phase separation in the cell at all expression levels (Fig. 2c; Fig. S3). Apart from tyrosine residues, ARID1A[PLD] primary sequence shows enrichment of glutamines with multiple polyQ tracts with stretches of three to four Gln residues in the C-terminal region (Tables S1, S2, S4). Previous studies have suggested that polyQ regions can promote LCD self-association[60,61]. However, when we mutated these Gln residues to Gly and created an ARID1A[PLD] variant, termed ARID1A[PLD 30QtoG], we observed only a modest (~2 fold) increase in intracellular $C_{sat}$ (Fig. 2c; Fig. S3). These data suggest that Tyr residues play a greater role in driving homotypic ARID1A[PLD] phase separation, similar to the FUS[PLD], whereas polar residues such as Gln play a moderate role.

Based on the results obtained from the sequence perturbations of FUS[2XPLD] and ARID1A[PLD], it appears that intracellular phase separation of PLDs can be tuned by the number of Tyr residues. Since BRG1[PLD], which only contains seven aromatic residues but a large number of proline residues, does not phase separate in cells, we attempted to improve its condensation driving force by increasing the tyrosine content. To this end, we created two variants where we mutated 17 proline and 41 proline residues to tyrosine residues, termed BRG1[PLD Aro+] and BRG1[PLD Aro++], respectively. We observed that both BRG1[PLD] variants can form intracellular condensates with comparable $C_{sat}$ to other mSWI/SNF PLDs (Fig. 2d; Fig. S3). However, the BRG1[PLD Aro++] was observed to form large irregular aggregates in the cytoplasm and was predominantly excluded from the nucleus (Fig. 2d; Fig. S3), which is likely due to strong homotypic interactions mediated by a large number of Tyr residues in this synthetic BRG1[PLD] variant.

## mSWI/SNF PLD condensates recruit low-complexity domains of transcriptional machinery and RNA polymerase II via heterotypic interactions

An emerging feature underlying transcriptional regulation by prion-like low complexity domains in transcription factors is their ability to directly engage with chromatin remodeler SWI/SNF complexes[34,40,45] and RNA polymerase II (RNA pol II)[25], the carboxy-terminal domain (CTD) of which also has a prion-like sequence (Fig. S5). Our previous studies have reported that BRG1, the catalytic subunit of the mSWI/SNF complex, can enrich within optogenetically induced FUS[PLD] (OptoFUS[PLD]) condensates and the FUS fusion protein, FUS-DDIT3, condensates in live cells[32]. We posited that this functional engagement can be, in part, mediated by disordered PLDs of FUS and BRG1. Indeed, FUS[2XPLD] condensates showed a strong colocalization with BRG1[PLD] in live cells (Fig. S6), which otherwise remains homogeneously distributed in the nucleus (Fig. 1f). This observation of BRG1[PLD] partitioning into FUS[2XPLD] condensates suggests the occurrence of heterotypic PLD-PLD interactions in these systems. To explore these heterotypic interactions systematically, we analyzed the degree of the partitioning of PLDs of FUS and RNA pol II (Table S4) into condensates formed by ARID1A[PLD], ARID1B[PLD,] and SS18[PLD]. To this end, we defined scaffolds and clients in each pair of PLD mixtures: scaffold is the protein that forms condensates ($C > C_{sat}$) and the client, defined as the protein that does not homotypically phase separate under the experimental conditions ($C < C_{sat}$), partitions into the scaffold condensates (Fig. 3a). The degree of client partitioning is determined by the sequence-specific scaffold-client interactions and the chain solvation free energy difference between the dense phase and the dilute phase[62–64]. When the experimental conditions are the same and the scaffold concentration is fixed, partition coefficients, defined as $k = I_{dense}/I_{dilute}$, of a group of similar clients to a scaffold condensate can report on the relative strength of scaffold-client interactions[62,63]. In our experiments, we observe that each of the PLD condensates can recruit other PLDs (Figs. 3b; S7), indicating a synergistic interplay between homotypic and heterotypic PLD interactions. However, the distribution of $k$ values (Fig. 3c) spans over two orders of magnitude

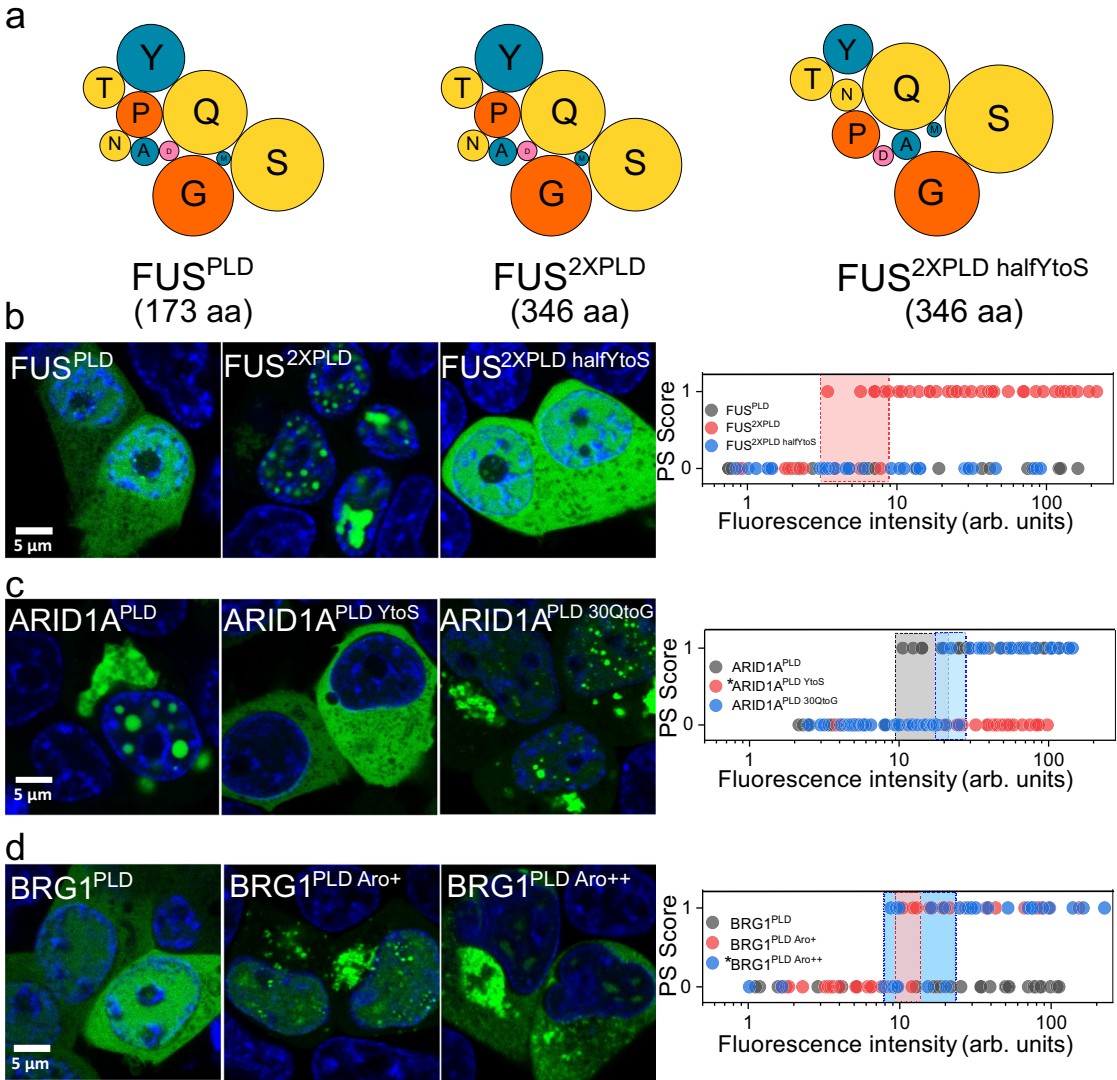

**Fig. 2 | Tyrosine residues play a dominant role in mSWI/SNF PLD phase separation. a** A bubble chart representation of the sequence composition of FUS$^{PLD}$, FUS$^{2XPLD}$, and FUS$^{2XPLD\ halfYtoS}$. The lengths of the PLDs are displayed as amino acid count "aa". Fluorescence microscopy images of HEK293T cells expressing GFP-tagged PLDs and variants of (**b**) FUS$^{PLD}$ (FUS$^{PLD}$, FUS$^{2XPLD}$, or FUS$^{2XPLD\ half\ YtoS}$), (**c**) ARID1A$^{PLD}$ (ARID1A$^{PLD}$, ARID1A$^{PLD\ YtoS}$ and ARID1A$^{PLD\ 30QtoG}$), and (**d**) BRG1$^{PLD}$ (BRG1$^{PLD}$, BRG1$^{PLD\ Aro+}$ and BRG1$^{PLD\ Aro++}$) as indicated. Hoechst was used to stain the cell nucleus, which is shown in blue. The phase separation capacity is quantified over various levels of nuclear protein concentrations. A phase separation (PS) score of '1' indicates the presence of nuclear condensates and a PS score of '0' represents diffused expression patterns. The shaded regions represent the transition concentrations. Asterisk '*' denotes cytoplasmic concentration. (FUS$^{PLD}$ $n = 25$ cells, FUS$^{2XPLD}$ $n = 51$ cells, FUS$^{2XPLD\ halfYtoS}$ $n = 32$ cells, ARID1A$^{PLD}$ $n = 29$ cells, ARID1A$^{PLD\ 30QtoG}$ $n = 49$ cells, BRG1$^{PLD}$ $n = 34$ cells, BRG1$^{PLD\ Aro+}$ $n = 38$ cells, and BRG1$^{PLD\ Aro++}$ $n = 28$ cells from two biological replicates). Also see Fig. S3.

(-2−200), suggesting that there is a broad range of specificity of heterotypic interactions. We observed a few common trends for all three scaffold condensates: SS18$^{PLD}$, ARID1A$^{PLD}$, and ARID1B$^{PLD}$. Firstly, the partitioning of three strongly phase-separating mSWI/SNF PLDs within each other's condensates are similar and comparable to their self-partitioning ($k$ ~ 50−220), suggesting that the sequence grammar driving homotypic PLD interactions are similar to the heterotypic interactions between these PLDs that governs co-partitioning. Secondly, BRG1$^{PLD}$, which showed a substantially lower propensity of phase separation (weaker homotypic interactions) than the other PLDs, partitioning within mSWI/SNF PLD condensates is almost 15-100-fold lower ($k$ ~ 2−5), suggesting substantially weaker heterotypic PLD interactions in this case. Interestingly, we observed that FUS$^{PLD}$ co-partitioned in mSWI/SNF subunit PLD condensates to a similar degree as the SS18$^{PLD}$, ARID1A$^{PLD}$, and ARID1B$^{PLD}$, ($k$ ~ 50−185) despite its $C_{sat}$ being an order of magnitude higher than mSWI/SNF subunit PLD condensates. These observations suggest highly favorable heterotypic

interactions, likely due to a homology in the sequence grammar of these PLDs. Finally, the CTD of RNA Pol II was observed to partition within mSWI/SNF PLD condensates to intermediate degrees ($k$ ~ 20−80; Fig. 3b, c; Fig. S7) as compared to other PLDs probed in this study.

Collective analysis of the experimental trends of partition coefficient data revealed a more positive correlation with the number of aromatic and arginine residues of the respective PLD chains (Fig. S8) than any other sequence features including the net charge per residue (NCPR), number of hydrophobic residues, and PLD chain length. This observation indicates that the aromatic and arginine residues may drive homotypic phase separation of the system as well as heterotypic PLD-PLD interactions leading to their co-partitioning[18,19,31]. However, we note that although aromatic and arginine residues showed the highest correlation ($R^2 = 0.59$) in our dataset, the overall low value of correlation suggests there exists a more complex interplay of interactions and chain solvation likely encoded by residues beyond these two residues.

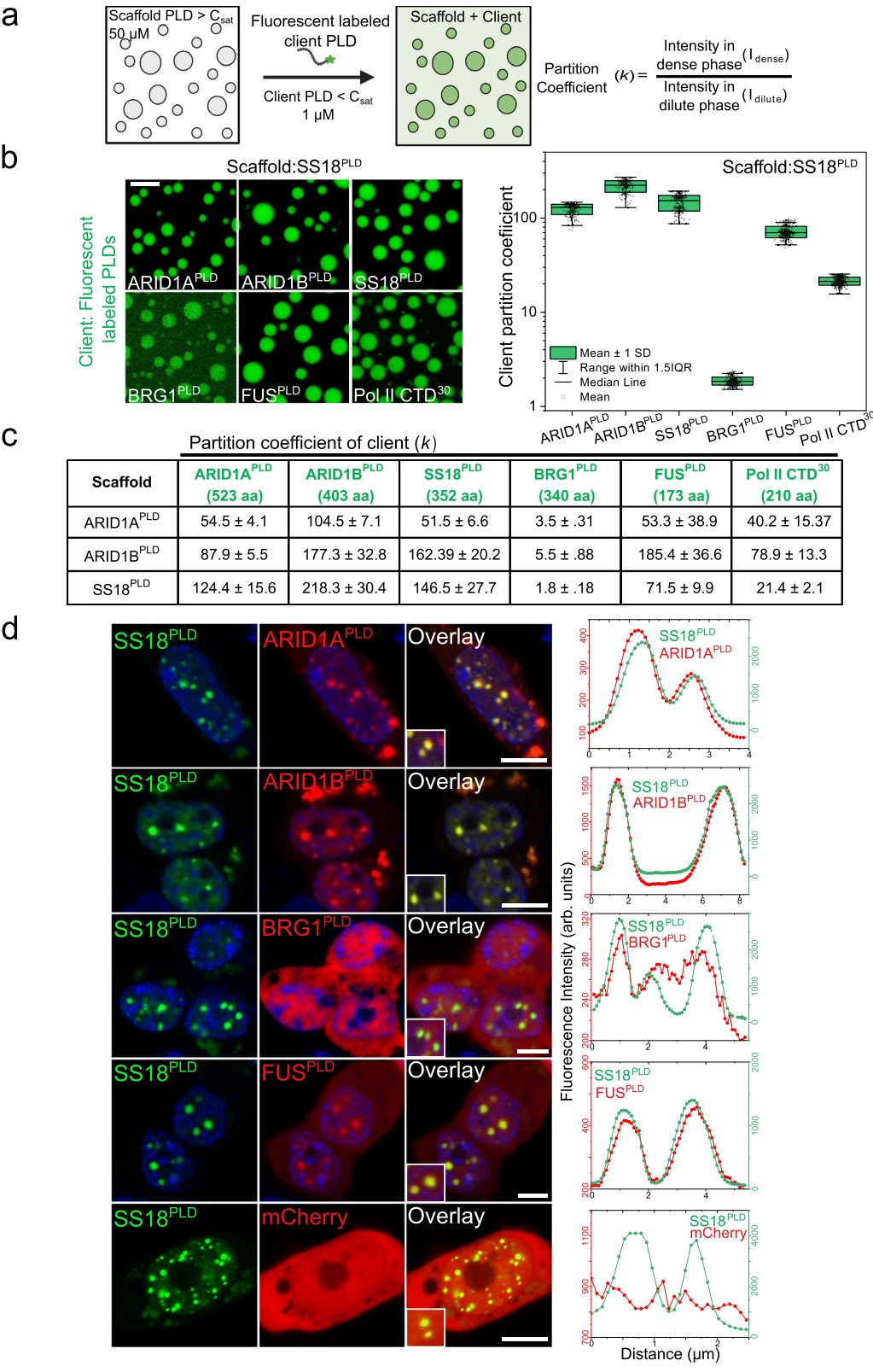

**Fig. 3 | Heterotypic PLDs interact and enrich within homotypic PLD condensates. a** A Schematic of the co-partitioning assay based on confocal fluorescence microscopy. 50 μM concentration of the scaffold$^{PLD}$ was used to form condensates and -1 μM of the AlexaFluor488 labeled client$^{PLD}$ was utilized to determine the partition coefficient ($k$). Created with BioRender.com. **b** Partitioning of AlexaFluor488 labeled client PLDs (ARID1A$^{PLD}$, ARID1B$^{PLD}$, SS18$^{PLD}$, BRG1$^{PLD}$, RNA Polymerase II CTD$^{30}$, and FUS$^{PLD}$) within condensates of SS18$^{PLD}$. Enrichment (partition coefficient) is calculated as shown in (**a**) and displayed as a box-and-whisker plot. (ARID1A$^{PLD}$ $n = 235$ condensates, ARID1B$^{PLD}$ $n = 222$ condensates, SS18$^{PLD}$

$n = 268$ condensates, BRG1$^{PLD}$ $n = 326$ condensates, RNA Polymerase II CTD$^{30}$ $n = 450$ condensates, and FUS$^{PLD}$ $n = 385$ condensates). The scale bar is 10 μm. **c** The average partition coefficient ($k$) is tabulated along with the standard deviation from the mean. **d** HEK293T cells co-expressing GFP-SS18$^{PLD}$ and either one of the mCherry-tagged PLDs (ARID1A$^{PLD}$, ARID1B$^{PLD}$, FUS$^{PLD}$ and BRG1$^{PLD}$) or mCherry alone. The degree of colocalization is displayed as intensity profiles for condensates shown in the inset images. Green represents the intensity profile of GFP-SS18$^{PLD}$ and red represents the profile for mCherry-tagged PLDs. The enrichment coefficients are reported in Fig. S9. The scale bar is 10 μm.

## Heterotypic interactions drive PLD co-condensation in live cells

Based on the observed extent of heterotypic PLD interactions among mSWI/SNF subunit PLDs and with FUS$^{PLD}$ in our in vitro co-partitioning assay (Fig. 3a−c), we next asked whether mSWI/SNF PLDs interact with each other and form co-condensates in living cells. To test this, we co-expressed pairs of PLDs with a GFP tag and a mCherry tag, respectively. In our first set of studies, we took advantage of the relatively low $C_{sat}$ of SS18$^{PLD}$ and considered it as the scaffold for live cell experiments. Heterotypic PLD interactions within phase-separated GFP-tagged SS18$^{PLD}$ condensates in live cells were probed by the degree of mCherry-tagged PLD co-partitioning. mCherry alone was used as a reference in our experiments. We observed that when expressed together, PLD mixtures formed heterotypic co-condensates in cellulo, despite their abilities to form homotypic condensates (Fig. 3d). This observation further supports the idea that there exists a substantial overlap between homotypic and heterotypic PLD interactions in these functionally-linked PLD systems. Quantification of the degree of enrichment revealed that while ARID1A$^{PLD}$ and ARID1B$^{PLD}$ are strongly colocalized with SS18$^{PLD}$ in condensates ($k > 2.0$), BRG1$^{PLD}$ only exhibited mildly enhanced enrichment ($k = 1.09$) compared to the mCherry control ($k = 0.99$), whereas FUS$^{PLD}$ showed a strong level of colocalization ($k = 1.53$; Fig. S9). This observed trend in our cellular assays is consistent with our results from in vitro partitioning experiments performed with purified proteins (Fig. 3b, c), further supporting that heterotypic PLD interactions are likely to be sequence-specific. We next performed similar experiments with GFP-tagged ARID1A$^{PLD}$ (Fig. S10) and ARID1B$^{PLD}$ (Fig. S11) as scaffold condensates and made similar observations that except for BRG1$^{PLD}$, other PLDs strongly co-localize together in the dense phase. However, we noted that Pol II CTD[30] did not significantly enrich within condensates of any of the three mSWI/SNF PLDs in cells (Fig. S12). This observation contrasts our in vitro client recruitment assay results showing strong enrichment of Pol II CTD[30] in mSWI/SNF PLD condensates (Fig. 3c). This may be due to post-translational modification of the Pol II CTD in cells, specifically, phosphorylation, which was previously shown to inhibit Pol II CTD recruitment to condensates formed by the FET family PLDs[65].

## Tyrosine residues are important for both homotypic and heterotypic interactions within mSWI/SNF PLD condensates

Results described in Fig. 2 above suggest that Tyr residues play an important role in driving homotypic phase separation of mSWI/SNF PLDs as well as the FUS$^{PLD}$. Due to the strong propensity of these distinct PLDs to colocalize in condensates, we next aimed to investigate the role of tyrosine residues in heterotypic PLD co-condensation. We first examined heterotypic interactions between FUS$^{PLD}$ and a variant of ARID1A$^{PLD}$ lacking tyrosine residues (Fig. 2c; Table S4). As discussed above, the wildtype ARID1A$^{PLD}$ condensates enrich FUS$^{PLD}$ in live cells (Fig. S10). However, when OptoFUS$^{PLD}$ was co-expressed with the 29Y-to-S variant of ARID1A (ARID1A$^{PLD\ YtoS}$), we observed that blue light-activated OptoFUS$^{PLD}$ condensates (see Materials and Methods) did not enrich ARID1A$^{PLD\ YtoS}$, suggesting a loss of heterotypic interactions (Fig. 4a). To test if increasing the Tyr content of a PLD chain can increase the degree of co-localization, we performed a similar experiment with two BRG1$^{PLD}$ variants containing 17 (BRG1$^{PLD\ Aro+}$) and 41 (BRG1$^{PLD\ Aro++}$) additional Tyr residues, respectively. We observe the BRG1$^{PLD\ Aro+}$ variant has almost 2-fold greater enrichment within the OptoFUS$^{PLD}$ condensates compared to the BRG1$^{PLD}$ (Fig. 4b, c). For BRG1$^{PLD\ Aro++}$ variant, however, we observed that the OptoFUS$^{PLD}$ condensates that formed on blue light activation did not show enrichment of BRG1$^{PLD\ Aro++}$, but pre-existing condensates of BRG1$^{PLD\ Aro++}$ were able to enrich OptoFUS$^{PLD}$ to a greater extent than WT BRG1$^{PLD}$ (Fig. 4b, c). Given BRG1$^{PLD\ Aro++}$ variant formed large irregular cytoplasmic aggregates in the cell, this may imply that in this case, the BRG1$^{PLD\ Aro++}$ homotypic interactions may have outweighed the heterotypic interactions, thereby driving the formation of discrete condensates similar to what was recently reported in the case of disordered FUS$^{PLD}$ and LAF1$^{RGG}$ systems[42]. It is also possible that BRG1$^{PLD\ Aro++}$ condensates are viscoelastic solids, and their formation has resulted in a near-complete depletion of the soluble fraction of this PLD. Taken together with the data shown in Fig. 2, these results suggest that tyrosine residues are important for both homotypic and heterotypic interactions among mSWI/SNF PLD and FUS$^{PLD}$ condensates.

## PLD condensates exhibit specificity in interactions with functionally-linked IDRs

The loss of heterotypic ARID1A$^{PLD}$ interactions with FUS$^{PLD}$ by 29Y-to-S mutations (Fig. 4a) suggests that there is a specificity of interactions among functionally-linked IDRs[66], which is likely to be encoded at the sequence level. To test the idea further, we attempted to probe interactions between PLD condensates with a naturally occurring non-prion-like IDR from a functionally unrelated protein. Since PLDs are predominantly associated with transcriptional activators and RNA-binding proteins[4], we reasoned to test if IDRs from a transcriptional repressor protein would enrich within PLD condensates. To this end, we chose the N-terminal IDR of FOXG1, which is a transcription factor predominantly acting as a transcriptional repressor in the developing brain[67]. The FOXG1 IDR contains 181 amino acids, comparable to FUS$^{PLD}$ (Fig. 5a), and has low aromaticity (Fig. 5b), and does not form condensates when overexpressed in cells (Fig. S13a). To probe interactions of FOXG1$^{N-IDR}$ with PLDs, we created condensates of recombinantly purified FUS$^{PLD}$ and SS18$^{PLD}$ and measured the enrichment coefficient of AlexaFluor488 labeled FOXG1$^{N-IDR}$. We observed that both condensates exclude FOXG1$^{N-IDR}$ (enrichment coefficient <1; Fig. 5c, d). In condensates of recombinant ARID1A$^{PLD}$ and ARID1B$^{PLD}$, there was a slight enrichment of FOXG1$^{N-IDR}$, but the partitioning was at least 2-fold lower than BRG1$^{PLD}$, which was the weakest PLD probed in this study in terms of the strength of heterotypic interactions (Fig. S13b, c). Based on the above results, we next tested if these observations hold in live cells by co-expressing FOXG1$^{N-IDR}$ with either SS18$^{PLD}$, ARID1A$^{PLD}$, ARID1B$^{PLD}$, or OptoFUS$^{PLD}$ condensates (Fig. 5e; Fig. S13d). In all cases, there was no enrichment of FOXG1$^{N-IDR}$ in any of the PLD condensates. Therefore, although the primary sequence of FOXG1$^{N-IDR}$ features similar spacer residues such as Pro, Gly, Gln, and Ser, partly similar to mSWI/SNF PLDs, it does not engage in heterotypic interactions with PLD condensates. Finally, to test if PLDs alone can be sufficient to explain the enrichment of a full-length mSWI/SNF subunit in FUS$^{PLD}$ condensates, we tested the folded domain of BRG1 for its ability to interact with FUS$^{PLD}$ condensates. We observed that the GFP-tagged BRG1$^{Folded}$ does not enrich within OptoFUS$^{PLD}$ condensates while both BRG1$^{PLD}$ and full-length BRG1 do (Fig. S14). These results collectively suggest the existence of a sequence grammar that encodes interactions among PLDs from functionally related proteins and between PLDs and non-prion-like IDRs of distinct sequence complexity as well as full-length proteins.

## mSWI/SNF PLDs lower the saturation concentration of FUS$^{PLD}$ and form spatially homogeneous co-condensates

How do heterotypic PLD interactions impact the phase behavior of multi-component PLD mixtures encompassing FUS and mSWI/SNF subunit PLDs? To systematically address this, we chose FUS$^{PLD}$ as our primary PLD system, which phase separates with a saturation concentration of ~ 200 μM in vitro (Fig. S2a), giving us a broad range of concentrations to test. To probe how mSWI/SNF PLDs affect FUS$^{PLD}$ phase separation, we chose two specific PLD systems: ARID1A$^{PLD}$, which features strong heterotypic interactions with FUS$^{PLD}$, and BRG1$^{PLD}$ that has weak heterotypic interactions with FUS$^{PLD}$ (Fig. 5d; Fig. S7). While FUS$^{PLD}$ ($C_{sat} \sim 200$ μM) has moderate phase separation driving force, BRG1$^{PLD}$ ($C_{sat} > 200$ μM) has a substantially lower tendency to undergo phase separation and ARID1A$^{PLD}$ ($C_{sat} \sim 2.5$ μM) is highly phase separation competent (Fig. 1; Fig. S2a). We observed that in the presence of

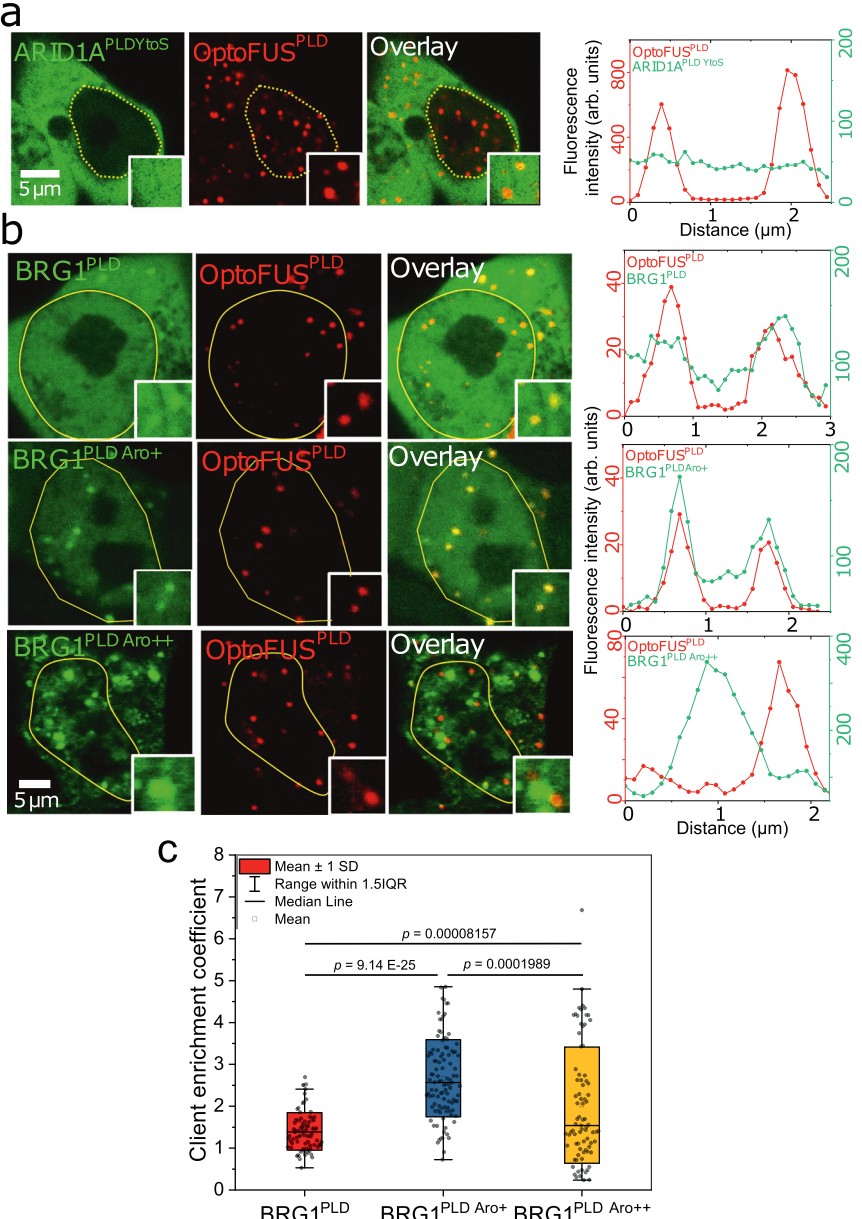

**Fig. 4 | Tyrosine residues are important for heterotypic interactions within mSWI/SNF PLD condensates. a** HEK293T cells co-expressing GFP-tagged ARID1A$^{PLD\ YtoS}$ and mCherry-tagged OptoFUS$^{PLD}$ constructs. The degree of colocalization is displayed as intensity profiles for condensates shown in the inset images. Green represents the intensity profile of the GFP-tagged construct and red represents the profile for mCherry-tagged construct. **b** HEK293T cells co-expressing PLDs (BRG1$^{PLD}$, BRG1$^{PLD\ Aro+}$, BRG1$^{PLD\ Aro++}$) and mCherry-tagged OptoFUS$^{PLD}$ construct. The degree of colocalization is displayed as intensity profiles for condensates shown in the inset images. Green represents the intensity profile of the GFP-tagged construct and red represents the profile for mCherry-tagged construct. The yellow line indicates the nuclear periphery. **c** Enrichment coefficients of GFP-tagged BRG1 PLDs within mCherry-tagged OptoFUS$^{PLD}$ condensates. Enrichment is calculated as the ratio of mean intensities from the dense phase and the dilute phase (BRG1$^{PLD}$ $n = 89$ condensates, BRG1$^{PLD\ Aro+}$ $n = 106$ condensates, BRG1$^{PLD\ Aro++}$ $n = 81$ condensates from two biological replicates). Significance was calculated by a student's two-tailed $t$ test.

both ARID1A$^{PLD}$ and BRG1$^{PLD}$ at concentrations below their respective $C_{sat}$, the saturation concentration of FUS$^{PLD}$ is lowered in a non-linear fashion (Fig. 6a; Fig. S15). These observations suggest that heterotypic interactions are highly cooperative with homotypic interactions in driving phase separation of the PLD mixtures[5]. In the co-PLD phase diagram (Fig. 6a), we identify five regimes: regimes (I) and (V) are homotypic phase separation regimes of the two PLDs; regime (II) is a single-phase regime where the mixture of PLDs stay soluble; regime (III) is a PLD co-condensation regime where each of the component PLD concentration is less than their respective $C_{sat}$ but the mixture undergoes phase separation, and regime (IV) is a PLD co-condensation

regime where each of the component PLD concentrations is higher than their respective $C_{sat}$.

Two emergent features of the two-component PLD phase diagrams (Fig. 6a; Fig. S15) are worth highlighting. The first of them is our observation that the PLD mixtures co-phase separate under conditions where the single PLD component concentrations are below their respective $C_{sat}$ (regime III). In biomolecular mixtures of multiple PLD components, the phase separation driving forces are effectively determined by the synergistic balance of homotypic and heterotypic PLD interactions. Intriguingly, the lowering of $C_{sat}$ of FUS$^{PLD}$ by mSWI/ SNF subunit PLDs (Fig. 6a; Fig. S15) shows a concave trend, which is

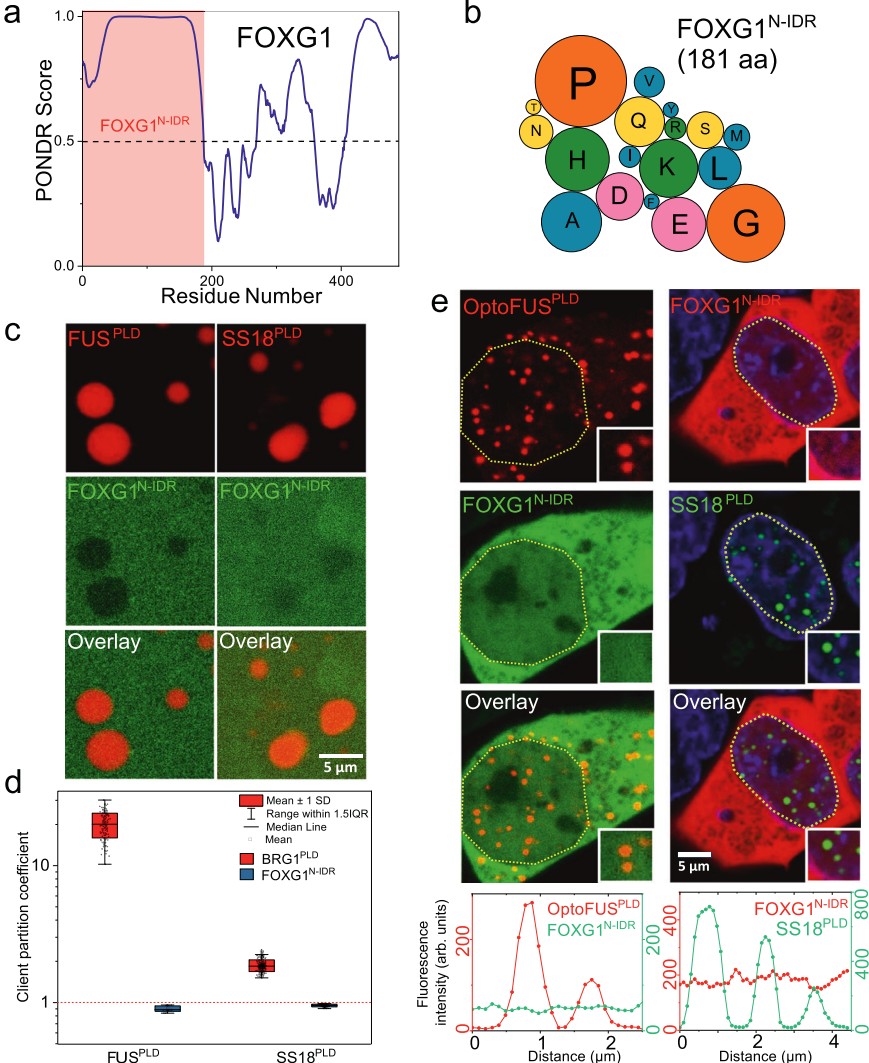

**Fig. 5 | Selectivity in IDR interactions with condensates formed by prion-like domains. a** The PONDR (Predictor of Natural Disordered Regions) score showing regions of disorder (>0.5) for the FOXG1 protein. The region shaded in red shows the N-terminus IDR for FOXG1 used in the study. **b** Amino acid composition of FOXG1$^{N-IDR}$. The color codes for amino acids are provided in Table S6. **c** Partitioning of AlexaFluor488 labeled FOXG1$^{N-IDR}$ within condensates of FUS$^{PLD}$ (250 µM) and SS18$^{PLD}$ (50 µM), respectively. **d** Enrichment is calculated as partition coefficient and displayed as a box-and-whisker plot for both FOXG1$^{N-IDR}$ and BRG1$^{PLD}$ (data

reported in Fig. 3b, c) within these condensates (FUS$^{PLD}$ condensates: FOXG1$^{N-IDR}$ $n = 37$, BRG1$^{PLD}$ $n = 153$ condensates; SS18$^{PLD}$ condensates; FOXG1$^{N-IDR}$ $n = 99$, BRG1$^{PLD}$ $n = 326$ condensates). **e** HEK293T cells co-expressing SS18$^{PLD}$ and mCherry-tagged FOXG1$^{N-IDR}$ or GFP-tagged FOXG1$^{N-IDR}$ and OptoFUS$^{PLD}$ constructs. The degree of colocalization is displayed as intensity profiles for condensates shown in the inset images. Green represents the intensity profile of GFP-tagged protein and red represents the intensity profile for mCherry-tagged protein.

indicative of positive cooperativity where the heterotypic interactions enhance the phase separation of the mixture[5]. Therefore, in this case, the heterotypic PLD interactions dominate over the homotypic interactions leading to an effective lowering of the $C_{sat}$ of either of the PLD chains.

The second key feature of the PLD mixture is the mixing of PLD dense phases within the co-condensates even though the $C_{sat}$ of individual PLDs differ by almost two orders of magnitude (Fig. 1; Fig. S2a). This is evident from confocal fluorescence microscopy images, which revealed that the PLD mixtures formed co-condensates that are spatially homogeneous in regimes III and IV (Fig. 6a–c). The mixing of individual PLD phases and the formation of PLD co-condensates (Fig. 6d) was not only observed in vitro but also in live cells when two PLDs were co-expressed (Fig. 3d; Figs. S10, S11). These observations again suggest that the sequence grammar driving homotypic and heterotypic PLD interactions are highly similar, leading to the dominance of heterotypic interactions in these mixtures (Fig. 6a; Fig. S15). If

the homotypic chain interactions were dominant over heterotypic interactions in the PLD mixtures, multi-phasic condensate morphologies may emerge[42,52,68,69] with spatially co-existing individual PLD phases instead of monophasic condensates.

## Heterotypic PLD interactions are detectable in the sub-saturation conditions

In our experiments thus far, we observed that condensates formed by mSWI/SNF subunit PLDs can recruit other PLDs in vitro and in live cells. Further, heterotypic PLD interactions lead to a lowering of FUS$^{PLD}$ saturation concentration and the formation of spatially homogeneous PLD co-condensates. Since heterotypic interactions seem to dominate over homotypic interactions in PLD mixtures (Fig. 6; Fig. S15), we next asked whether these PLDs interact with each other at concentrations below their saturation concentrations and if their interactions in the dilute phase are also governed by the LCD primary sequence features. To address this, we first employed a bead halo assay[70] with a pair of

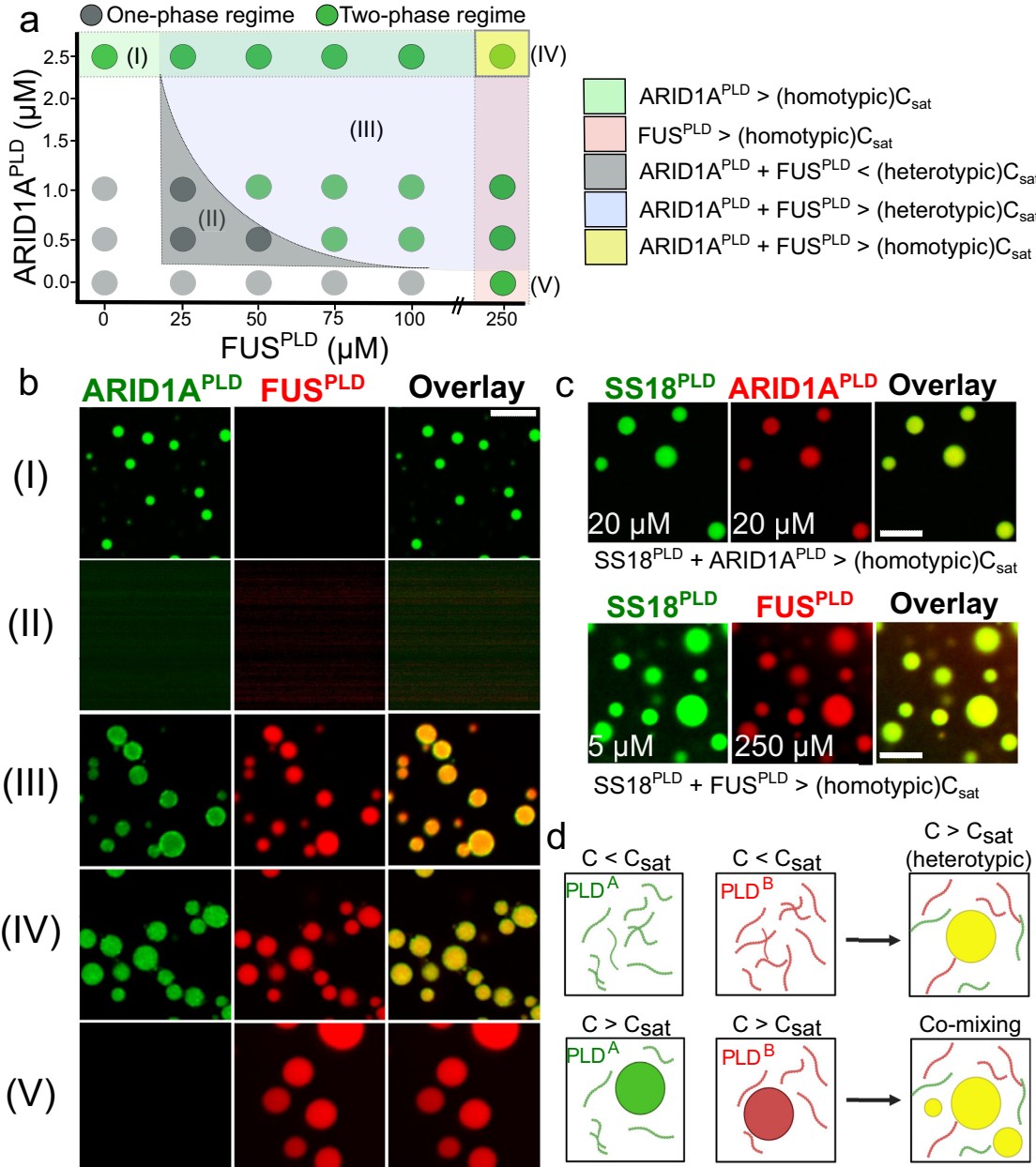

**Fig. 6 | Heterotypic PLD interactions promote phase separation and form monophasic PLD co-condensates. a** Co-phase diagram of FUS$^{PLD}$ and ARID1A$^{PLD}$ shows lowering of saturation concentration of heterotypic PLD mixtures. The green circles indicate two-phase regime, and the gray circles indicate single-phase regime. The legend describes the shaded regions highlighted in five distinct colors. **b** Fluorescence microscopy images of PLD samples from the specific regions of the phase diagram shown in (**a**). ARID1A$^{PLD}$ is labeled with AlexaFluor488 and FUS$^{PLD}$ is labeled with AlexaFluor594. **c** Fluorescence microscopy images of PLD co-

condensates formed by the mixtures of SS18$^{PLD}$ with ARID1A$^{PLD}$ (top) and SS18$^{PLD}$ with FUS$^{PLD}$ (bottom) at concentrations above the saturation concentrations of respective PLDs. SS18$^{PLD}$ is labeled with AlexaFluor488, ARID1A$^{PLD}$, and FUS$^{PLD}$ are labeled with AlexaFluor594, respectively. The scale bar is 5 μm for all images. **d** A schematic showing the formation of monophasic PLD co-condensates, corresponding to regimes III and IV in the phase diagram shown in (**a**). This is created with BioRender.com.

PLDs. In these experiments, AlexaFluor488-labeled scaffold PLDs containing a hexahistidine (His$_6$) tag were immobilized on the surface of Ni-NTA micro-spheres through Ni-His$_6$ interactions. The scaffold PLD was designed to contain a solubility tag (MBP; see Materials and Methods) to abrogate homotypic phase separation on the bead surface. We used two mSWI/SNF PLDs, SS18$^{PLD}$ and BRG1$^{PLD}$, as scaffolds in our experiments. The bulk scaffold PLD concentration used for these measurements was fixed at 250 nM, which is much lower than their respective $C_{sat}$. As a negative control, we used an AlexaFluor488-labeled His$_6$-MBP containing a short linker peptide (GGGCGGG) without any PLDs. Next, 250 nM of AlexaFluor594-labeled FUS$^{PLD}$ (client

PLD) was added to the solution (Fig. 7a). We expect that if heterotypic PLD interactions are present at these scaffold-client concentrations that are much lower than their $C_{sat}$, they would enable the recruitment of the client PLD (FUS$^{PLD}$) to the bead surfaces coated with a scaffold PLD (SS18$^{PLD}$ or BRG1$^{PLD}$). Further, the relative degree of client recruitment will depend on the relative strength of scaffold-client interactions under these conditions. Indeed, we observed that FUS$^{PLD}$ is preferentially recruited on the SS18$^{PLD}$- and BRG1$^{PLD}$-coated beads whereas beads coated with MBP-alone did not show such client enrichment (Fig. 7b, c). These data suggest that SS18$^{PLD}$ and BRG1$^{PLD}$ interact with FUS$^{PLD}$ at a concentration much lower than their

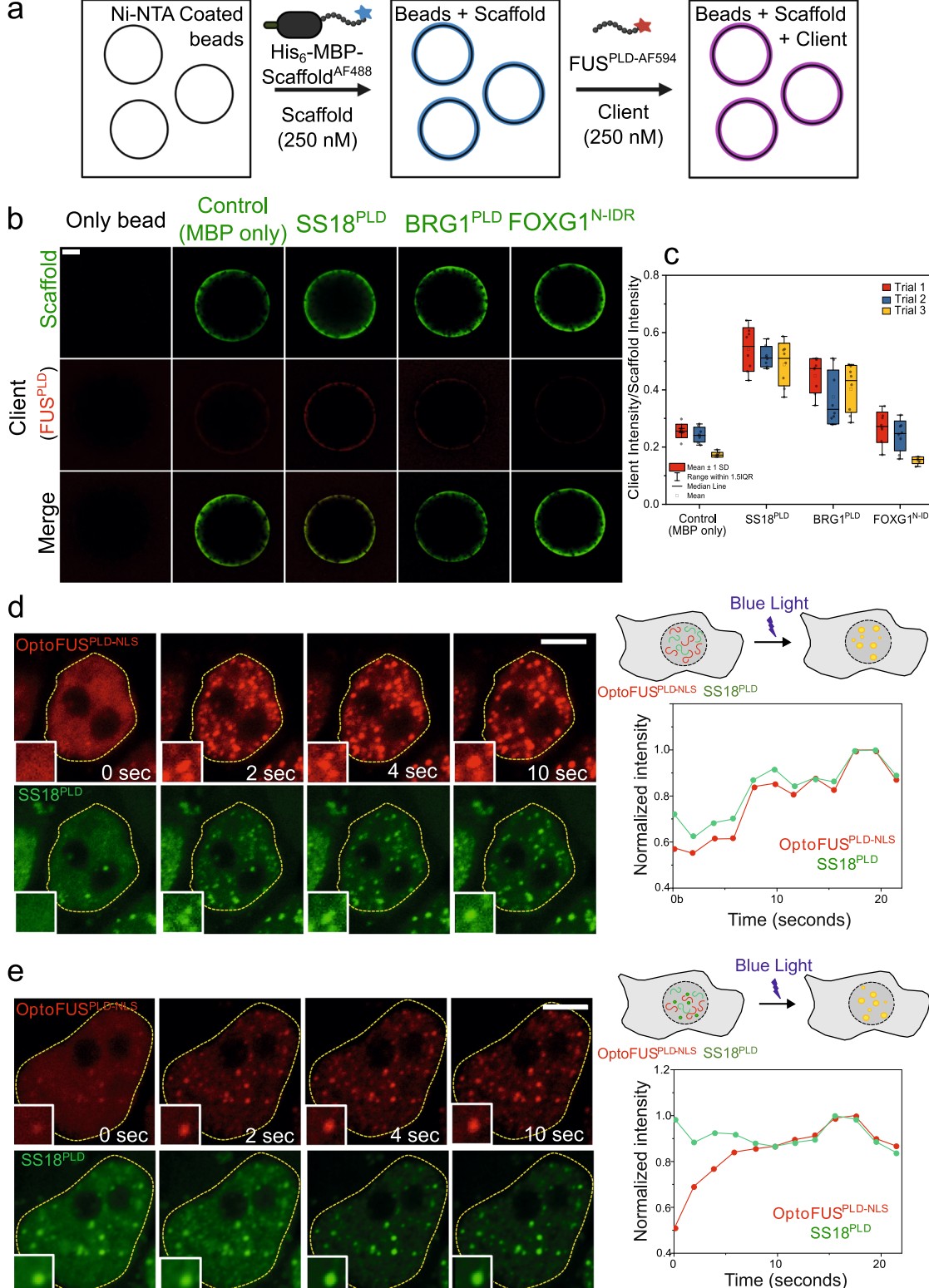

**Fig. 7 | Heterotypic PLDs interact at sub-saturation concentrations. a** A schematic representation of the bead halo assay. Created with BioRender.com.
**b** 250 nM of AlexaFluor488-labeled His₆-MBP-Scaffold constructs — where scaffold signifies either SS18^PLD, BRG1^PLD, FOXG1^N-IDR, or control His₆-MBP-GGGCGGG, were added to Ni-NTA beads. 250 nM of FUS^PLD labeled with AlexaFluor594 was then added to the above beads (see Materials and Methods for further details). Binding was quantified using the ratio of fluorescence intensities (fluorescence signal from the client FUS^PLD/fluorescence signal from the scaffold) on the surface of the bead.
**c** A box-and-whisker chart of the intensity ratios is plotted with the mean and standard deviation (*n* = 8 beads/trial). Significance is shown in Fig. S16.

**d** HEK293T cells co-expressing OptoFUS^PLD-NLS (Cry2-mCherry-FUS^PLD-NLS) and GFP-SS18^PLD below their saturation concentrations. Upon blue light activation, OptoFUS^PLD co-condenses with GFP-SS18^PLD. Mean intensity profiles of the co-condensates formed are shown as a function of time for condensates within the inset image. Green represents the intensity profile of GFP-SS18^PLD and red represents the intensity profile for OptoFUS^PLD. The corresponding movie is shown in supplementary movie 1. **e** Pre-existing GFP-SS18^PLD clusters act as nucleation sites for OptoFUS^PLD condensates upon blue light activation (also see supplementary movie 2 and supplementary movie 3). The scale bar is 5 μm for all images.

saturation concentrations. We further noted that the relative FUS$^{PLD}$ enrichment was much higher on SS18$^{PLD}$-coated beads as compared to BRG1$^{PLD}$-coated beads (Fig. 7c; Fig. S16), suggesting that SS18$^{PLD}$ is a significantly stronger scaffold than the BRG1$^{PLD}$. We also compared these results with a non-prion-like IDR, FOXG1$^{N-IDR}$, which was found to be non-interactive with PLD condensates in vitro and in live cells (Fig. 5). Similar to the MBP control, FOXG1$^{N-IDR}$-coated beads failed to recruit FUS$^{PLD}$. These results not only provide evidence of the presence of heterotypic PLD-PLD interactions at their sub-saturation concentrations but also lend further support that the heterotypic interactions between FUS$^{PLD}$ and mSWI/SNF component PLDs are sequence-specific (Fig. 7c). We also attempted to quantify apparent binding affinities by titrating FUS$^{PLD}$ concentration and keeping the scaffold PLD concentration fixed. However, instead of an apparent two-state binding isotherm, we observed a monotonic increase in the enrichment of FUS$^{PLD}$ on the bead surface (Fig. S17) as the FUS$^{PLD}$ concentration increased. This observation may suggest non-stoichiometric binding and homotypic interactions between the FUS$^{PLD}$ chains on the surface of the bead at higher client concentrations[71,72].

The occurrence of nanoscopic homo-oligomers of FUS in the pre-phase separation regime has recently been reported[43]. Our observations that heterotypic PLDs interact and recruit each other on the bead surface at concentrations much lower than their saturation concentrations now suggest that heterotypic interactions between prion-like LCDs can also occur in the dilute phase independent of phase separation. However, a key unanswered question is whether such interactions can be observed in the complex intracellular microenvironment of a living cell. To test heterotypic interactions between mSWI/SNF subunit PLDs and FUS$^{PLD}$ in the absence of phase separation, we employed a light-activated phase separation approach[53]. In this assay, we used an OptoFUS$^{PLD-NLS}$ construct to induce FUS$^{PLD}$ condensation in live cell nucleus using blue light, while co-expressing GFP-SS18$^{PLD}$ near sub-saturation level (Fig. 7d). When FUS$^{PLD}$ condensation was actuated by blue light, we made two key observations. The first one was that SS18$^{PLD}$ was enriched simultaneously at sites where OptoFUS$^{PLD-NLS}$ condensates were formed upon blue light activation (Fig. 7d; Supplementary Movie 1). We note that in this case, there were no pre-existing SS18$^{PLD}$ condensates. Secondly, we observed that in cells containing pre-existing SS18$^{PLD}$ clusters, they acted as nucleation centers for OptoFUS$^{PLD}$ condensation (Fig. 7e; Supplementary Movie 2). This feature is further highlighted in cells with multiple pre-existing SS18$^{PLD}$ condensates where OptoFUS$^{PLD}$ condensation predominantly occurred at those sites (Fig. S18; Supplementary Movie 3). Collectively, we conclude that FUS and SS18 PLDs form soluble heterotypic complexes in the dilute phase below their saturation concentrations and SS18$^{PLD}$ clusters can nucleate condensation of the FUS$^{PLD}$ in live cells (Figs. 7d, e; S18).

## Discussion

mSWI/SNF (also known as BAF) complex is a multi-subunit ATP-dependent chromatin remodeler with critical functions in genome organization and spatiotemporal transcriptional programming during development[47,48,73,74]. Mutations in mSWI/SNF subunits including ARID1A/B, BRG1, and SS18 are linked to multiple tumor types. However, apart from the ATP-dependent catalytic activity of the subunit BRG1 in nucleosome repositioning and eviction[75,76], the functions of other subunits in controlling chromatin landscape are less understood. Interestingly, one common feature among ARID1A/B, BRG1, and SS18 primary sequences is that they all have long stretches (~300–500 amino acids) of disordered low-complexity domains without any known functions[49]. Sequence analysis revealed that these LCDs are prion-like (Fig. S1)[32]. Employing in vitro experiments with purified LCDs as well as cell culture models, here we show that ARID1A/B and SS18 PLDs undergo phase separation with saturation concentrations

substantially lower than previously reported PLDs from RNA-binding proteins, including FUS$^{PLD}$ (Fig. 1; Fig. S2). We hypothesized that similar to the PLDs from FET families[26,27], the tyrosine residues provide a major driving force for their high phase-separation capacity. We tested this idea by creating 29Tyr-to-Ser variant of ARID1A$^{PLD}$ and 17Pro-to-Tyr and 41Pro-to-Tyr variants of BRG1$^{PLD}$ (Fig. 2c, d). Mutations of Tyr residues abolished phase separation propensity of ARID1A$^{PLD}$ completely in cells, similar to mutating half the tyrosine residues of FUS$^{2XPLD}$ to serine that resulted in a complete loss of phase separation (Fig. 2b), whereas introducing Tyr residues to BRG1$^{PLD}$ increased its propensity to form intracellular condensates significantly (Fig. 2d). We also made a 30Gln-to-Gly variants of ARID1A$^{PLD}$, which only increased the $C_{sat}$ of ARID1A$^{PLD}$ ~ 2-fold. Together, these results suggest that Tyr residues are a key determinant of mSWI/SNF PLD phase separation, whereas polar residues such as Gln plays a modest role.

Not only in homotypic interactions, we found that Tyr residues play dominant roles in heterotypic PLD interactions between the mSWI/SNF PLDs and FUS$^{PLD}$. The key roles of aromatic amino acids in mSWI/SNF PLD phase separation and engagement with FUS$^{PLD}$ may stem from the ability of Tyr residues to interact with multiple other residues including arginine, proline, and methionine in addition to aromatic residues[31,77–81]. Since mSWI/SNF subunit PLDs are enriched with proline and methionine residues (Tables S1–3; Fig. 1b), the presence of aromatic amino acids can therefore magnify the number of multivalent contacts between PLD chains. This is further supported by the fact that inclusion of 17 Tyr residues to BRG1$^{PLD}$ was sufficient to lower its intracellular saturation concentration to a level comparable to other mSWI/SNF PLDs (Fig. 2d). While FUS$^{PLD}$ and FUS$^{2XPLD\ halfYtoS}$ with 24 aromatic residues did not phase separate in cells, BRG1$^{PLD\ Aro+}$, which contains 24 aromatic residues, was able to form intracellular condensates at much lower expression level (Fig. 2). Our results, therefore, are consistent with a model for PLD phase separation where Tyr residues play a key role by mediating a diverse set of inter-chain interactions. Similarly, Tyr-mediated interactions are likely to play an equally important role in driving heterotypic PLD associations (Fig. 4b, c; Fig. S8), driving specificity in partner recruitment (Fig. 5), and formation of PLD co-condensates (Fig. 6). However, we also acknowledge that interactions mediated by non-aromatic residues can play a sizable role in shaping the overall phase behavior of the PLD systems and uncovering them in the future studies will lead to a more complete picture of the underlying molecular grammar.

The saturation concentrations of the PLDs of SS18, ARID1A/B, and FUS$^{PLD}$ differ by two orders of magnitude at room temperature (Fig. 1; Fig. S2), yet mSWI/SNF PLD condensates formed completely miscible co-condensates with FUS$^{PLD}$ in vitro and in live cells (Figs. 3 and 6). This observation seems puzzling at first based on the difference in $C_{sat}$ values of respective PLD chains, which may represent highly dissimilar strengths of homotypic inter-chain interactions in FUS$^{PLD}$ and mSWI/SNF PLDs. If the homotypic interactions are substantially different in a mixture of IDRs, they can form coexisting dense phases with differential densities[42,52,68,69]. This is likely the case for BRG1$^{PLD\ Aro++}$ variant, which formed discrete condensed phases when co-expressed with OptoFUS$^{PLD}$ condensates (Fig. 4b, c). However, except for this synthetic variant, our co-phase diagrams in vitro (Fig. 6; Fig. S15) and PLD co-condensation results in live cells suggest that heterotypic interactions between mSWI/SNF PLD chains are dominant over the homotypic interactions in PLD mixtures. This assertion is consistent with the observed miscibility of FUS$^{PLD}$ condensates with ARID1A$^{PLD}$ and SS18$^{PLD}$ condensates. The observed positive cooperativity in interactions between PLDs of mSWI/SNF subunits and that of FET proteins may have important functional relevance in the formation of transcriptional hubs where transcription factors and coactivators can coexist in a single homogeneous phase-separated hub through co-scaffolding (Fig. 8). This implies that multiple proteins with PLDs can provide a

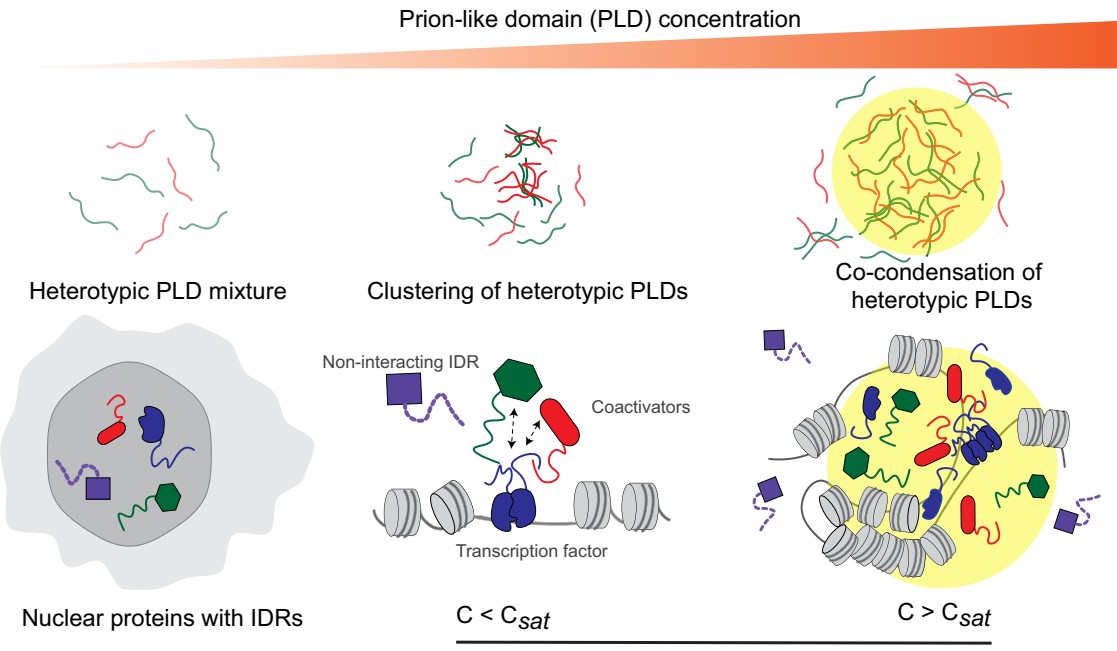

Prion-like domain (PLD) concentration

Heterotypic PLD mixture

Clustering of heterotypic PLDs

Co-condensation of heterotypic PLDs

Nuclear proteins with IDRs

Non-interacting IDR

Coactivators

Transcription factor

$C < C_{sat}$

$C > C_{sat}$

Assembly of transcriptional machinery

**Fig. 8 | Schematic illustration of heterotypic PLD-mediated co-assemblies driving functional protein interaction networks.** Heterotypic PLD interactions occur at sub-saturation concentrations which upon increasing protein concentration can lead to co-phase separation of the mixture into spatially homogeneous multi-component condensates (top panel). Our results collectively suggest that transcriptional proteins can be assembled into co-phase-separated hubs through sequence-specific positively cooperative interactions among low-complexity domains (bottom panel).

positive cooperative effect to reduce the concentration required for phase separation of the collection of proteins. We speculate that heterotypic PLD-mediated positive cooperativity in protein-protein interactions is likely to play key roles in the operation of SWI/SNF complexes and their interactions with transcription factors containing similar low-complexity domains. Interestingly, FOXG1[N-IDR], a disordered non-prion-like domain of a transcriptional repressor, was observed to be excluded from PLD condensates in cells or has a very low enrichment coefficient in vitro (Fig. 5; Fig. S13). These data suggest a complex sequence grammar that incorporates specificity within PLDs for partner interactions, which is likely to be important for their biological functions.

The ability of the FET family of transcription factors containing an N-terminal PLD to orchestrate oncogenic gene expression has recently been linked to their aberrant interactions with the BAF complex subunits, such as BRG1. Previously, heterotypic protein-protein interactions have been reported for transcription factors containing PLDs such as EBF1 and FUS[34], as well as FET proteins and FUS-DDIT3[33,82,83]. Given multiple mSWI/SNF subunits contain long PLDs, could such interactions be mediated by these intrinsically disordered LCDs (Fig. 8)? Indeed, our results suggest that mSWI/SNF subunit PLDs can engage in sequence-specific interactions with each other and with the PLD of FUS. We also observe that FUS-DDIT3 form co-condensates with mSWI/SNF PLDs (Fig. S19). Moreover, the relative degree of PLD partitioning into FUS-DDIT3 condensates followed a similar trend as PLD-only condensates (Fig. S19b; Fig. S9a). Since PLDs are common in many endogenous and cancer-specific fusion transcription factors, based on our results reported here, we speculate a common mode of functional protein-protein networking in transcriptional regulation for these factors through sequence-specific heterotypic interactions between low-complexity domains.

Finally, a key finding of our study is that the PLD-mediated multivalent interactions can occur at sub-micromolar concentrations below their saturation concentrations (Fig. 7). These results imply that heterotypic PLD-mediated protein-protein interactions are likely to be present at physiologically relevant protein concentrations inside living cells. Such interactions can lead to the formation of heterotypic clusters at the single-phase regime, similar to homotypic pre-percolation clusters observed for RNA-binding proteins[43]. Given intracellular concentrations of many proteins at their endogenous level often resides at sub-saturation level[84], our results reported in this work may suggest that the functional protein-protein interaction networks can be mediated by multivalent LCDs independent of phase separation.

## Methods

### Protein expression, purification, and labeling
A list of proteins used in the study is provided in Table S4 along with their amino acid sequences. Codon-optimized proteins used in this work were gene-synthesized by GenScript USA Inc. (Piscataway, NJ, USA) and cloned into pET His6 MBP N10 TEV LIC cloning vector (2C-T) [was a gift from Scott Gradia (Addgene plasmid # 29706)]. Proteins were expressed, purified, and fluorescently labeled as described in our earlier work[32]. FOXG1[N-IDR] was expressed in BL21-CodonPlus (DE3)-RIPL competent cells and was purified using the same protocol as other constructs. All recombinant proteins contained three exogenous amino acids (SNI) at their N-termini after TEV cleavage.

### Cell culture
The HEK293T cells (a kind gift from Drs. Jae Lee and Soo Lee at University at Buffalo, SUNY) were cultured at 37 °C and 5% $CO_2$ in Dulbecco's Modified Eagle's Medium (Gibco™ 11965092) supplemented with 10% fetal bovine serum (Gibco™ A3160501). To initiate transfection, 20,000 cells were seeded in Nunc™ Lab-Tek™ II chambered coverglass (8 wells). After 24 h, Lipofectamine 2000 reagent (Thermofisher 11668030) was used to transfect the cells with 0.5 μg plasmid, according to the manufacturer's protocol. The cells were imaged after 20−24 h of transfection. Colocalization experiments were carried out with 0.5 μg plasmid for each construct. Table S5 provides a complete list of plasmids used for protein expression in HEK293T cells.

## Fluorescence imaging

To facilitate live-cell imaging, the cells were moved to FluoroBrite DMEM Media (Thermofisher A1896701) containing 10% FBS and Hoechst33342 dye (Thermofisher H3570; 1 µg/ml) one hour before the imaging. Imaging was carried out using either a Zeiss LSM710 laser scanning confocal microscope (Plan-Apochromat 63x/1.4 oil DIC M27) or a Q2 laser scanning confocal microscope (ISS Inc., 63X objective), with the cells being maintained at 37 °C on a temperature and CO2-controlled stage. For light activated $FUS^{PLD}$ condensation in live cells (opto$FUS^{PLD}$), the Cry2 homo-oligomerization domain (previously characterized by Shin et al.[53]) was fused to $FUS^{PLD}$ or $FUS^{PLD-NLS}$ containing a nuclear localization signal (NLS): PAAKRVKLD[85]. These constructs also contained an mCherry tag for live cell fluorescence imaging. Opto$FUS^{PLD}$/Opto$FUS^{PLD-NLS}$ droplets were formed by exposing the cells to blue light (488 nm) for a minute during imaging. Image processing was carried out using FIJI and CellProfiler[86,87]. For tracking Opto$FUS^{PLD-NLS}$ droplets, peak detection was performed using NumPy in SciPy and droplets were tracked using Trackpy[88].

## In vitro phase separation experiments

The proteins were buffer exchanged into a 25 mM Tris-HCl buffer (pH 7.5) containing 125 mM NaCl at room temperature. After preparing the samples at the desired protein concentrations, TEV protease (TEV: protein = 1:25 v/v) was added and the mixture was incubated for 1 h at 30 °C to cleave the $His_6$-MBP-N10 tag. 10-20% Ficoll PM70 was used as a crowder for some samples as mentioned in the appropriate data figures. Next, 4 µl of the sample was placed in the center of a microscope glass slide that was fitted with a custom-made containment, created using the broad end of a plastic pipette tip, and sealed onto the slide. The top of the chamber was then sealed with parafilm to prevent evaporation, and the samples were incubated for 45−60 min at room temperature. Finally, the samples were imaged using a Zeiss Primovert inverted iLED microscope (40x objective) or a Zeiss LSM710 laser scanning confocal microscope (Plan-Apochromat 63x/1.4 oil DIC M27). Microscopy images were recorded and processed using ZEN (blue, v2.3) for Zeiss Primovert and ZEN (SP5 2012 Black) for Zeiss LSM710.

## Partition coefficient analysis

For partition coefficient analysis, either a Zeiss LSM710 laser scanning confocal microscope (Plan-Apochromat 63x/1.4 oil DIC M27) or a Q2 laser scanning confocal microscope (ISS Inc., 63X objective) was used to record condensate images. Phase-separated condensates were prepared following the procedure outlined above with a trace amount (~1−2%) of fluorescently labeled proteins. Confocal images were collected 1 h after sample preparation. Using CellProfiler, droplets were segmented and the mean intensity within each droplet ($I_{dense}$) was determined. Additionally, for each image, five spots outside the condensates were randomly selected to estimate the background signal, and the mean intensity of the external dilute phase was calculated ($I_{dilute}$). Finally, the partition coefficient ($k$) was calculated by taking the ratio of the two intensities ($k = I_{dense}/I_{dilute}$). For determining the enrichment coefficients in cells, images were imported to CellProfiler, and GFP-PLD condensates were segmented. After segmentation, the mean intensity of the GFP signal and the mean intensity of the mCherry signal within the condensates were obtained. Five to eight spots within the nucleus surrounding the GFP-PLD condensates were randomly selected to estimate the background signal. Enrichment coefficient analysis was then performed as above.

## Estimation of saturation concentration for PLDs in cells

Images of cells expressing GFP-tagged PLDs or variants were captured using a laser scanning confocal microscope (Q2-ISS Inc., 63X objective) with fixed imaging parameters across samples. Images were imported into CellProfiler[86], and nuclei of the cells were segmented using Hoechst intensity. After segmentation, mean GFP intensity was obtained within the segmented region as a proxy for either nuclear or cytoplasmic protein concentration. Each cell was then manually examined for the presence or absence of condensates.

## Fluorescent recovery after photobleaching (FRAP) analysis

For FRAP experiments, a circular region of interest was bleached with 100% power for ~1−2 s which was followed by an imaging scan for 60 s. The recorded AlexaFluor488-labeled probe intensity or GFP intensity values from the bleached ROI were then corrected for photofading by normalizing them to an unbleached reference condensate. Multiple FRAP recovery curves were averaged for each sample and plotted as a function of time using OriginPro (2018b). The dimension of the bleaching ROI was constant across samples. For each time point of the FRAP recovery curve, the standard deviation of the intensity values from the three or more FRAP curves was taken as the uncertainty.

## Bead Halo Assay (BHA)

20 µl of HisPur™ Ni-NTA Magnetic Beads (Thermo Fisher Scientific 88831) slurry was resuspended in 480 µl of buffer (25 mM Tris-HCl, pH 7.5, 125 mM NaCl) and mixed by inversion of the tube. The beads were pulled down at the side of the tube using a magnetic stand. The supernatant was removed, and the wash step was repeated two additional times. The beads were finally resuspended in 200 µl buffer, which was subsequently used as a working stock solution. For the experimental setup, 1 µl of beads was diluted in 3 µl of buffer and 0.5 µl of 2.5 µM AlexaFluor488-labeled scaffold. These proteins contained a hexahistidine ($His_6$) tag, which enabled their immobilization on the bead surface via Ni-$His_6$ interactions. The scaffolds also contained an MBP solubility tag to prevent their surface condensation[89]. The scaffold-coated beads were allowed to equilibrate for 15 minutes, following which 0.5 µl of 2.5 µM of AlexaFluor594-labeled client ($FUS^{PLD}$) was added to achieve a final concentration of 250 nM of both the scaffold PLD and the client PLD. The samples were incubated for 15 minutes and imaged using a laser scanning confocal microscope (Q2 laser scanning microscope, 63X objective). For estimating the relative enrichment of $FUS^{PLD}$ on the scaffold-coated beads, a ratio of the mean fluorescence intensity of the client (AlexaFluor594) on the bead to the mean fluorescence intensity of the scaffold (AlexaFluor488) was taken and normalized to the labeling efficiency. This normalized data was subsequently defined as the client enrichment score and compared across samples (Fig. 7c). For our attempt to estimate a binding curve for heterotypic PLD complexation, the client ($FUS^{PLD}$) concentration was continuously varied as indicated in Fig. S17, keeping the scaffold concentration fixed at 250 nM.

## Statistics and reproducibility

The sample size, number of trials, and statistical methods used for each experiment and data analysis have been provided in the respective figure captions. Line intensity profiles in Figs. 3d, 4a, 5e, 6b, c, 7d, e and in the supplementary data figures are representative images of at least two independent trials.

## Reporting summary

Further information on research design is available in the Nature Portfolio Reporting Summary linked to this article.

# Data availability

All data relevant to the findings of this manuscript are included in the manuscript and the Supplementary Information file. Source data for all graphs are provided in the Source Data file. Source data are provided with this paper.

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

## Acknowledgements

This work was supported by the US National Institute of General Medical Sciences (NIGMS) of the National Institutes of Health grant award (R35 GM138186) to P.R.B. The initial part of this work was supported by the US National Institute on Aging (NIA) of the National Institutes of Health grant award (R21 AG064258) to P.R.B. The authors gratefully acknowledge the members of the Banerjee Lab, Dr. Rohit Pappu and Dr. Tanja Mittag for valuable feedback during the manuscript preparation.

## Author contributions

Conceptualization: P.R.B. and R.B.D.; Methodology: P.R.B., R.B.D., A.S. and A.K.R.; Investigation: P.R.B., R.B.D., M.M.M., and A.K.R.; Resources: P.R.B.; Writing – original draft: P.R.B., M.M.M. and R.B.D.; Writing – reviewing and editing: P.R.B. and R.B.D.; Funding acquisition: P.R.B.

## Competing interests

P.R.B. is a member of the Biophysics Reviews (AIP Publishing) editorial board. This affiliation did not influence the work reported here. All other authors declare no competing interests.
