## [Peer Review File · Nature Communications]

REVIEWER COMMENTS

Reviewer #1 (Remarks to the Author):

In Davis et al., the authors use a bead halo assay to measure heterotypic interactions between the PLDs of FUS and either SS18 PLD or BRG1PLD at concentrations where they do not form condensates. This is important because interactions under these conditions are more likely to be biologically relevant, based on their concentrations in normal cells. The choice of a bead halo assay, the controls, and methods are all appropriate to the question. I did have one concern: For each condition, the authors measure the fluorescence intensity ratios on 8 beads; It is unclear whether these bead images are from the same or multiple technical replicates. In Figure 6, the ratio of fluorescence intensity is for FUS/SS18 is ~ 0.3 for 250nM of FUS, while in Figure S11 the ratio at that same concentration is < 0.02 . Is this indicative of the level of variability in the assay or other differences between those experiments? Either the methods should clarify existing technical replicates, or they should be performed.

Provided the pattern of SS18PLD > BRG1PLD > GFP holds across replicates, the results support the authors conclusion that SS18PLD and BRG1PLD interact the dilute phase, not just in condensates.

Minor comments:

In figure 6b, the scale bar is partially obstructed.

Reviewer #2 (Remarks to the Author):

Davis et al. extend their previous study (<https://onlinelibrary.wiley.com/doi/full/10.1002/pro.4127>) on the co-condensation of prion-like FET proteins and the SWI/SNF complex. The previous study already established the role of heterotypic interactions in driving the co-condensation of PLDs with the condensates formed by one of these proteins. This paper extends these findings a bit, but more importantly, proposes the idea that heterotypic interactions in the dilute phase also exist. I am somewhat puzzled by this finding. How else would heterotypic interactions operate to bring the protein chains together in the condensed phase? Is there a proposal in the literature that the interactions get activated when the droplets form? Given the close correspondence with previously published results by

the same authors, I am skeptical that Nature Communications is a proper venue for these results. But the editors can make the appropriate decision in this regard.

I have several other major and minor comments that the authors may find helpful.

(i) While reading the introduction, it was a bit difficult to put everything in the context of the broader field. The authors appear to be mostly focusing on the narrow singular view encoded in the stickers-spacers model, which has been shown to be inconsistent with numerous experimental studies. I expect a paper to be more balanced, at least in the introduction, in capturing the current state-of-the-art in the field, even if the authors buy into a specific philosophy more or provide rigorous evidence within the paper for believing as such. This should involve mutagenesis experiments involving residues of different types, and not just perceived stickers.

(ii) I find the presentation of the part about chain length lacking in many respects. Surely, one can design longer chains that phase separate poorly. This can happen if the composition of stickers is less than in shorter chains. If I look at the composition of ARID1A, Y and R residues are significantly lower than other residues. What is the driving force here? Is the phase separation abolished if Y or R residues are mutated to spacer residues such as L?

(iii) The cooperative co-condensation of two PLDs below C_{sat} is almost presented as a new result. There are many examples in the literature where this has been shown previously. An extreme example of such behavior is the complex coacervation between two oppositely charged proteins.

(iv) I find the paper making many conclusions or general statements that are not partially or fully supported by the preceding data. In other cases, these statements are not new or unexpected but presented as such. This makes it difficult to put the results of this paper in proper context.

(v) The authors may want to develop a simple theoretical model that takes into account the dilute phase binding affinity between the same or different PLDs and how changing it will change the C_{sat} . The paper only presents the expected behavior in a highly qualitative manner. The only way to make it interesting is to start the quantification of such effects that the condensate field is lacking in many respects.

(vi) I kept getting confused with the phrase "ternary PLD mixture" as many of the figures appear to show results from a binary mixture.

Reviewer #3 (Remarks to the Author):

The manuscript by Davis et al. "Heterotypic interactions in the dilute phase can drive co-condensation of prion-like low-complexity domains of FET proteins and mammalian SWI/SNF complex" is an interesting and well-written manuscript providing evidence that the PLD domains are important for interactions between chromatin remodeling components (SWI/SNF) and RNA-binding FET proteins. The latter are known to form recurrent fusion proteins in around 20 cancer types leading to aberrant transcription. This manuscript is therefore relevant for broad areas such as chromatin remodeling complexes and their interaction with transcription factors, transcription and when these processes are dysregulated in cancer.

Davis et al. show that the "strength" of the PLD is sequence-dependent, mainly determined by the number of stickers (aromatic residues and arginine) which determine both phase separation capacity and co-partitioning. They note that heterotypic PLD interactions are very important, and in many cases stronger than homotypic interactions, which leads to complete mixing in the condensates and, as expected, lowering of saturation concentrations. For the first time, they also relate the PLD interactions observed in subsaturated concentrations in the dilute phase directly with co-condensation when condensation is induced. This is important since much debate is ongoing whether phase separation is required for many of biological processes such as transcriptional activation or if increased interactions of PLD proteins are enough. The manuscript is generally very well-written and the figures are clear. However, some concerns must be addressed. Specifically which parts of the PLD are really required for the interactions/phase separation? They show that the PLD is enough for interaction and condensate formation, but do relevant proteins such as SWI/SNF components without a PLD interact with and co-partition into condensates? Another option would be to infer mutations in the PLD such as exchanging tyrosines with serines and evaluate if interactions are lost.

Major remarks:

1. If you want to make strong conclusions regarding phase separation capacity of FET proteins and that PLDs are important for SWI/SNF assembly, some more experiments are needed (Figure 1C/F) to strengthen these claims, namely, to include EWSR1 and/or TAF15, and SMARCC1/2 in these analyses. SMARCC1/2 are very important for the assembly of the SWI/SNF core complex.
- 2a. When you introduce how to study condensates in live cells, you describe that these PLDs could form condensates at low expression levels, and then say that you have done transient transfection. You need to be careful how to write about transient transfection and expression levels. According to my experience, transient transfection does not lead to low expression values, and they rarely match endogenous levels. Please look over the text with this in mind and perhaps include in discussion.
- 2b. In figure 1F-G, I thought it was nice that you show that different expression levels ("concentrations") of PLDs lead to either diffuse pattern or condensate formation. However, it would be interesting to see if condensates in the transition region forms smaller condensates than at larger concentrations. What

concentration is the image from? For example, at endogenous expression levels, FUS-DDIT3 form many small condensates while overexpressed FUS-DDIT3-EGFP form few, large condensates (Owen et al., 2021). As pointed out, the level of PLDs is very important and overexpression changes their properties and perhaps function.

2c. A minor remark, in Figure 1F-G. Is a random number of condensates included for the different PLDs? Please also include number of replicates (n) in figure legends for all experiments.

3. An interesting result was how the length, or really the number of aromatic and arginine stickers, determines the phase separation capacity. This was described nicely in the text and discussion. However, the title of section 2 (The length of PLD chains determines their saturation concentration in live cells; line 189) is a bit misleading. Consider revising and substituting the word determines. Also, the same applies to line 456 in the discussion (strong phase separation driving forces arise from their longer chain lengths). Another option would be to investigate whether the same amount of aromatic and arginine stickers in PLDs of different lengths show similar capacity for condensate formation.

4a. In the scaffold/client experiments in figure 3B, it would be nice to have microscopy images of the condensates before the client PLD is added. Also, it would be very valuable to have a negative control, preferably a SWI/SNF component without a PLD domain, to show that not “everything” is partitioning into the scaffold condensates, i.e. that partitioning is due to the PLD domain.

4b. The same is true in Figure 6B, D, E. It would strengthen your results to again include another negative control, such as SWI/SNF component without PLD for the bead halo assay to show that these are not enriched at the bead surface. For Figure 6D, E, a negative control is needed to show that only specific proteins co-localize directly when FUS condensates are induced.

5. In the discussion, the second to last section raises an interesting question (line 509) but there is a lack in the reasoning in the paragraph. On line 511, please make it clear if you refer to heterotypic interactions of the SWI/SNF complex. Then on line 513, what do you mean with “this idea”? I also do not understand why you introduce “new” experiments in the discussion and how the result of this experiment helps the conclusion further. Also, it has been known for at least 10 years that FUS has a diffuse pattern but co-localizes to FUS-DDIT3 condensates when the fusion is present (Owen et al., 2021; Thelin-Järnum, Göransson, Burguete, Olofsson, & Åman, 2002; Thomsen, Grundevik, Elias, Ståhlberg, & Åman, 2013). Instead, if you want to continue these discussion points and draw conclusions about FET fusion proteins, it would be much more worthwhile to perform more experiments with FUS-DDIT3-EGFP and other SWI/SNF PLDs, to determine their co-partitioning in condensates, and if the partition coefficient of FUS and FUS-DDIT3 differ. Although keep in mind that the FET proteins are present in live cells.

6. One of the major concepts of this manuscript is the importance of the sequence-specificity versus the number of sticker amino acids. For example, the different co-partitioning of FUSPLD and BRG1PLD. Are they explained by other differences than the number of stickers? In addition, the bad phase competence of BRG1PLD, is it only due to the lower amount of aromatic/arginine stickers? Furthermore, is it possible to separate sequence-specificity with phase separation capacity when it comes to PLD interactions? Please clarify and expand the discussion on these questions.

Minor remarks:

7a. The text is generally clear and well-written, but paragraph 3 in the introduction can be improved such as the last sentence (line 81-83). Also on line 76, clarify what you mean with rich phase behavior.

7b. In the discussion, end of paragraph 3 and paragraph 4 should be rewritten for clarity. I think it would be good to merge these sections since they overlap in content.

8. In Figure 1B, what does the color-coding mean? If it is based on amino acid properties, that would be nice, but this need to be clear from the figure or at least the figure legend.

9. Based on the FRAP in Figure 1E, is it really correct to write that all PLDs, including ARID1B, are rapidly exchanging molecules (line 140-141)?

10. In result part 3 (line 223), the headline is misleading, you have not investigated any transcription factors in this section. FUS is not in itself a transcription factor, only in the oncogenic fusion form such as FUS-DDIT3.

11a. The enrichment coefficients from Figure 4A/S6 do not seem to match the images, for example the amount of SS18 that partitions into condensate for ARID1A and 1B looks different in the images. Are representative cells shown? Please indicate in all figure legends how many cells/condensates are used. Please clarify.

11b. In Figure 4A/Figure S7/S8, I do not understand how you know that the scaffold protein labeled with GFP is the one forming the condensates since we know both PLDs cooperate with forming the condensates. Clarify this in the text.

11c. In some instances, in Figure 4A/Figure S7/S8, the scaffold protein (ARID1A, ARID1B) has more diffuse pattern when co-expressed with BRG1. Why? Does this point to interactions in the dilute phase? Or do BRG1 interfere with the scaffold protein in the condensates? Are these representative images? The same trend is seen in Figure S9 when co-expressed with Pol II. Here ARID1B do not seem to form condensates in the nuclei. These questions should be addressed.

12. Please explain what you mean with a ternary PLD mixture, the first time mentioned in introduction and/or results. Also clarify what you mean with the following sentence or rephrase, "co-mixing of individual PLD phases within ternary PLD condensates" (line 346-347).

13. In the discussion, you describe that FUS and SWI/SNF PLD have positive cooperativity. Do you know of any examples of PLD-PLD interactions that do not have positive cooperativity (line 488) and what does that lead to? Consider discussing.

References:

Owen, I., Yee, D., Wyne, H., Perdikari, T. M., Johnson, V., Smyth, J., . . . Shewmaker, F. (2021). The oncogenic transcription factor FUS-CHOP can undergo nuclear liquid-liquid phase separation. *Journal of Cell Science*, 134(17). doi:10.1242/jcs.258578

Thelin-Järnum, S., Göransson, M., Burguete, A. S., Olofsson, A., & Åman, P. (2002). The myxoid liposarcoma specific TLS-chop fusion protein localizes to nuclear structures distinct from PML nuclear bodies. *International Journal of Cancer*, 97(4), 446-450. doi:10.1002/ijc.1632

Thomsen, C., Grundevik, P., Elias, P., Ståhlberg, A., & Åman, P. (2013). A conserved N-terminal motif is required for complex formation between FUS, EWSR1, TAF15 and their oncogenic fusion proteins. *FASEB Journal*, 27(12), 4965-4974.

We thank all the reviewers for their critical feedback which has helped us to improve the clarity of the present work significantly. Following their critiques and our due deliberation, we have added a substantial amount of additional new experimental results in the revised manuscript that not only strengthened the conclusions made in the original version of our manuscript but also provided new insights into the physical principles that govern sequence-specific heterotypic PLD interactions and their co-condensation. Below we provide a point-by-point response to the reviewers' comments (our responses are in blue). **Major changes/additions of new materials** in the main text and the SI are highlighted in yellow.

Reviewer #1 (Remarks to the Author):

In Davis et al., the authors use a bead halo assay to measure heterotypic interactions between the PLDs of FUS and either SS18 PLD or BRG1PLD at concentrations where they do not form condensates. This is important because interactions under these conditions are more likely to be biologically relevant, based on their concentrations in normal cells. The choice of a bead halo assay, the controls, and methods are all appropriate to the question. I did have one concern: For each condition, the authors measure the fluorescence intensity ratios on 8 beads; It is unclear whether these bead images are from the same or multiple technical replicates. In Figure 6, the ratio of fluorescence intensity is for FUS/SS18 is ~0.3 for 250nM of FUS, while in Figure S11 the ratio at that same concentration is <0.02. Is this indicative of the level of variability in the assay or other differences between those experiments? Either the methods should clarify existing technical replicates, or they should be performed. Provided the pattern of SS18PLD > BRG1PLD > GFP holds across replicates, the results support the authors conclusion that SS18PLD and BRG1PLD interact the dilute phase, not just in condensates.

We thank the reviewer for the critique and for recognizing that our bead halo assay conditions probe biologically relevant interactions between functionally linked PLDs under sub-micromolar concentrations. In the revised manuscript, results from three technical replicates for the bead halo assay are provided. In each trial, a fresh set of beads was coated with the scaffold PLD or the control and used for these experiments. In each trial, we probed a set of intensity ratios for 8 beads. These results are now included as updated in **Fig. 7b-c (provided below)**. Clearly, the pattern SS18^{PLD} > BRG1^{PLD} > MBP control holds across replicates.

In addition, we have also added appropriate controls to further support our claims. For the negative control, we used an AlexaFluor488-labeled His₆-MBP (maltose binding protein) containing a short linker peptide (GGGCGGG) without any PLDs. The single cysteine residue was used to site-specifically label it with AlexaFluor488. This new construct ensured that we had the same fluorophore between different PLD samples and the control (earlier, we used GFP as a negative control). We have also included a new biologically relevant control, the N-terminal non-prion-like IDR of FOXG1 (FOXG1^{N-IDR}), which we have now shown to be a non-interactor of FUS^{PLD} and SS18^{PLD} using *in vitro* and *in cellulo* experiments (**Fig. 5**). Similar to the MBP control, FOXG1^{N-IDR}-coated beads failed to recruit FUS^{PLD}. These results not only provide evidence of the presence of heterotypic PLD-PLD interactions at their sub-saturation concentrations but also lend further support that the heterotypic interactions between FUS^{PLD} and mSWI/SNF component PLDs are sequence-specific.

Together, with three biological replicates and the inclusion of appropriate controls, our bead halo assay results suggest the following pattern of interaction strengths $SS18^{PLD} > BRG1^{PLD} > FOXG1^{N-IDR} \sim \text{Control (His}_6\text{-MBP only)}$ with FUS^{PLD} . These updated results are shown below.

Figure 7: Heterotypic PLDs interact at sub-saturation concentrations. a) A schematic representation of the bead halo assay. Created with BioRender.com. b) 250 nM of AlexaFluor488-labeled His₆-MBP-constructs - SS18^{PLD}, BRG1^{PLD}, FOXG1^{N-IDR}, or control His₆-MBP was attached to Ni-NTA beads. 250 nM of FUS^{PLD} labeled with AlexaFluor594 was then added to the above beads (see Materials and Methods for further details). Binding was quantified using the ratio of fluorescence intensities (fluorescence signal from FUS^{PLD}/fluorescence signal from the scaffold) on the surface of the bead. c) A box-and-whisker chart of the intensity ratios is plotted with the mean and standard deviation (n = 8 beads/ trial). Significance is shown in Fig. S16.

Figure S16: Statistical analysis for the bead halo assay data shown in Fig. 6a-c in the main text. Three trials were combined (with points for each of the trials shown in different colors) to calculate the significance using Student's t-test. **** P ≤ 0.00001. Here the control refers to the AlexaFluor488-labeled His₆-MBP containing a short linker peptide (GGGCGG) without any PLDs.

We would like to point out that the apparent discrepancies in **Fig. 7** and **Fig. S11 (Fig. S17 in the revised SI)** noted by the reviewer are due to different intensities of laser excitation used for imaging AlexaFluor594-FUS^{PLD} in different experiments aimed at addressing independent questions. As one can imagine, the *Client Intensity/Scaffold Intensity ratio* depends on the intensity of the 594 nm and 488 nm laser lines used in a given set of experiments. For the measurements reported in **Fig. 7c**, these excitation laser intensities were kept fixed across samples. However, in **Fig. S17**, we titrated the FUS^{PLD} (client) concentration up to 10-fold (2.5 μ M) as compared to **Fig. 7** (FUS^{PLD} concentration = 250 nM). Hence, in **Fig. S17**, we used a much lower excitation intensity for the 594 laser line as compared to **Fig. 7**, while keeping the excitation intensity for the 488 nm laser line the same. The 594 nm laser line intensity was kept constant across samples with variable FUS^{PLD} (client) concentration. The idea behind the results shown in **Fig. S17** was to check if we could extract binding constants between the two PLDs (scaffold: SS18 and client: FUS), which turned out not to be the case due to non-stoichiometric binding and homotypic interactions between the FUS^{PLD} chains on the surface of the bead at higher client concentrations. **Since these are independent experiments aimed at probing a different question than what is reported in Fig. 7**, we did not find a reason to maintain the same experimental settings and conditions during imaging. In summary, the data reported in Fig. 7 with new controls hold independent of the data reported in Fig. S17. We hope that our explanation now will clear up this confusion.

Minor comments:

In figure 6b, the scale bar is partially obstructed.

We have updated **Fig. 6b** (now **Fig. 7b** in the revised manuscript).

Reviewer #2 (Remarks to the Author):

Davis et al. extend their previous study (<https://onlinelibrary.wiley.com/doi/full/10.1002/pro.4127>) on the co-condensation of prion-like FET proteins and the SWI/SNF complex. The previous study already established the role of heterotypic interactions in driving the co-condensation of PLDs with the condensates formed by one of these proteins. This paper extends these findings a bit, but more importantly, proposes the idea that heterotypic interactions in the dilute phase also exist. I am somewhat puzzled by this finding. How else would heterotypic interactions operate to bring the protein chains together in the condensed phase? Is there a proposal in the literature that the interactions get activated when the droplets form? Given the close correspondence with previously published results by the same authors, I am skeptical that Nature Communications is a proper venue for these results. But the editors can make the appropriate decision in this regard.

We are surprised that this Reviewer finds our paper merely an extension of our previous study. Based on the presentation of our results, although we thought it would be clear to the readers that what we report here are significant new findings beyond what was reported in 2021 by Davis et al. (<https://doi.org/10.1002/pro.4127>), we realize that it may not be the case, so below we provide a brief account of the new findings reported in this manuscript.

- For the first time, we show that prion-like domains of multiple mSWI/SNF subunits form dynamic phase-separated condensates in live cells.

- Using live cell-based experiments, we show that tyrosine residues play a dominant role in mSWI/SNF subunit PLD phase separation similar to the FUS^{PLD}, whereas polar residues such as Gln play a moderate role.
- Using in vitro and live cell-based experiments, we show that heterotypic interactions drive PLD co-condensation in live cells. These experiments clearly show that tyrosine residues are important for both homotypic and heterotypic interactions within mSWI/SNF PLD condensates.
- Using in vitro and live cell-based experiments, we show the existence of a sequence grammar that encodes interactions among PLDs from functionally related proteins and between PLDs and non-prion-like IDR of distinct sequence complexity as well as full-length proteins. (newly added to the revised manuscript)
- We show that the sequence grammar driving homotypic and heterotypic PLD interactions are highly similar, leading to the dominance of heterotypic interactions in these mixtures even though the C_{sat} of individual PLDs differ by almost two orders of magnitude.
- Using in vitro and live cell-based experiments, we show, for the first time that Heterotypic PLD interactions are detectable in the sub-saturation conditions. *“this is important because interactions under these conditions are more likely to be biologically relevant, based on their concentrations in normal cells.”* The reviewer-1 noted during the first round of review while describing these results. Methodologically, we employ bead halo assay in vitro and a light-inducible phase separation assay in live cells to detect sequence-specific complexations between LCDs at sub-saturation concentrations. We envision that these methodologies will be broadly useful in the condensate community to study interactions between low-complexity domains of different sequence grammars.

Whether our results are worthy of being published in *Nature Communications* or not is a judgment call, which will depend on the editor and collectively all the reviewers. Overall, we are a bit surprised by the dismissive tone of this reviewer starting with the way we wrote our introduction. We respectfully note that the reviewer gives short shrift to the current work reported in this manuscript. We argue that the combination of experimental results collected for five PLDs in vitro and in live cells (**a total of twenty-nine constructs were probed in live cells and twelve constructs were purified for in vitro studies**), in isolation and in mixtures, are pretty significant efforts in the field. However, it is the reviewer’s prerogative to make such assertions.

The Reviewer did raise some major concerns, which we have tried very hard to address. These are summarized below.

I have several other major and minor comments that the authors may find helpful.

(i) While reading the introduction, it was a bit difficult to put everything in the context of the broader field. The authors appear to be mostly focusing on the narrow singular view encoded in the stickers-spacers model, which has been shown to be inconsistent with numerous experimental studies. I expect a paper to be more balanced, at least in the introduction, in capturing the current state-of-the-art in the field, even if the authors buy into a specific philosophy more or provide rigorous evidence within the paper for believing as such. This should involve mutagenesis experiments involving residues of different types, and not just perceived stickers.

There is no “buying into any specific philosophies”. The Stickers and spacers remain arguably one of the best polymer models to explain **the phase behavior of PLD chains**. In addition to the existing pieces of literature that are cited in our manuscript (Ref # 18-27 in the revised manuscript), we note that a recent large-scale computational study on PLDs by Joseph and

colleagues furthers the utility of this model for describing PLD phase behavior (Ref # 28; Maristany, J., et al, bioRxiv 2023.06.14.543914 (2023) doi:10.1101/2023.06.14.543914). The reviewer said many studies have provided experimental evidence that the sticker-and-spacer model does not work, but did not specify where we can find these studies. We have looked very hard but did not find such reports. However, we note a few key studies from Mittal and co-workers on different systems than PLDs (LAF1 RGG and synthetic IDPs) where they rightly argue that “multiple residues” matter for phase separation beyond the aromatic stickers (Rekhi, S. et al. bioRxiv 2023.03.02.530853 (2023). doi:10.1101/2023.03.02.530853; Schuster, B. S., et al. (2020) PNAS, 117(21), 11421–11431. <https://doi.org/10.1073/pnas.2000223117>). **We note that the LAF1 RGG domain and synthetic IDP systems that were the subject of these studies have distinct sequence grammars than PLD chains**, which are largely devoid of charged residues. We agree that in a broader context, the sequence grammar encoding LCD phase separation can be more complex and additional factors beyond aromaticity, such as the net charge and hydrophobicity of LCDs are likely to play equally dominant roles in their phase separation. However, we wanted to emphasize the sequence features of prion-like LCDs that drive their phase separation in our discussion, given PLDs are the central focus of the present study. Based on this point, we posit that an exciting future study could be experimentally and computationally focused on these distinct LCDs and their systematic variants and compare the performance of the existing polymer models including the sticker-and-spacer model, which is clearly beyond the scope of this current study.

We have now revised the introduction to capture these differences between PLDs and other associative LCDs with distinct sequence grammars and cited these papers by Mittal and colleagues of LAF1 RGG and synthetic IDP systems.

Further to this point; in the revised manuscript, we have also included new experimental data with new constructs to show that the Tyr residues play a dominant role in mSWI/SNF subunit PLD phase separation similar to the FUS^{PLD}, whereas polar residues such as Gln play a moderate role. In addition, using in vitro and live cell-based experiments, we show that heterotypic interactions drive PLD co-condensation in live cells. These experiments clearly show that tyrosine residues are important for both homotypic and heterotypic interactions within mSWI/SNF PLD condensates. The details of these experiments and results are provided in the response to the next point below. However, in the discussion section of the revised manuscript, we also note that *interactions mediated by non-aromatic residues can play a sizable role in shaping the overall phase behavior of the system and uncovering them in future studies will lead to a more complete picture of the underlying molecular grammar.*

(ii) I find the presentation of the part about chain length lacking in many respects. Surely, one can design longer chains that phase separate poorly. This can happen if the composition of stickers is less than in shorter chains. If I look at the composition of ARID1A, Y and R residues are significantly lower than other residues. What is the driving force here? Is the phase separation abolished if Y or R residues are mutated to spacer residues such as L?

We thank the reviewer for raising this point. We have revised our discussions on the chain length in the revised manuscript and replaced them with the results and discussions on the PLD sequence features that drive the phase separation and heterotypic co-condensation. Specifically, we have now included new data with new constructs to show that Tyr residues play a key role in the driving force for phase separation of the mSWI/SNF PLDs tested in the system. However, non-aromatic polar residues, such as glutamines also play a modest role. To this end, we have created five new constructs.

(1) A new variant of FUS^{2XPLD} where we substituted Tyr residues from the 2nd half of the FUS-PLD with Ser residues (FUS^{2XPLD halfYtoS}). Unlike the FUS^{2XPLD}, which phase separates in live cells at a relatively low expression level, FUS^{2XPLD halfYtoS} remained diffused at an order of magnitude higher expression level, similar to FUS^{PLD}. This is remarkable since FUS^{2XPLD halfYtoS} has twice the length of FUS-PLD and the only difference between this variant and FUS^{2XPLD} is that the former contains 24 Tyr residues and the latter contains 48 Tyr residues. This data suggests that the length alone is insufficient to drive phase separation of a PLD chain, it requires a critical number of aromatic residues in this case.

(2) We have made 29YtoS mutations in ARID1A^{PLD}, which abolished the phase separation capacity of AIRD1A^{PLD} in HEK293T cells, again suggesting the role of Tyr residues in driving phase separation in this system.

(3) We have made 30QtoG mutations in ARID1A^{PLD}, which increased the saturation concentration for phase separation of AIRD1A^{PLD} in HEK293T cells by ~ 2 fold, suggesting a modest role of non-aromatic residues in driving phase separation in this system.

(4) We have introduced 17 and 41 additional Tyr residues in WT-BRG1^{PLD} in two new synthetic variants, which we named BRG1^{PLD ARO+} and BRG1^{PLD ARO++}, respectively. Although WT-BRG1^{PLD} does not phase separate in the cell at any expression level, both BRG1^{PLD ARO+} and BRG1^{PLD ARO++} showed intracellular condensates, with BRG1^{PLD ARO++} showing predominantly large irregular clusters in the cytoplasm.

These new data have been added as **Fig. 2** and **Fig. S4** in the revised manuscript and the corresponding text and discussion have been modified/added.

Figure 2: Tyrosine residues play a dominant role in mSWI/SNF PLD phase separation. a) A bubble chart representation of the sequence composition of FUS^{PLD}, FUS^{2XPLD}, and FUS^{2XPLD half YtoS}. The lengths of the PLDs are displayed as amino acid count “aa”. Fluorescence microscopy images of HEK293T cells expressing GFP-tagged PLDs and variants of b) FUS^{PLD} (FUS^{PLD}, FUS^{2XPLD}, or FUS^{2XPLD half YtoS}), c) ARID1A^{PLD} (ARID1A^{PLD}, ARID1A^{PLD YtoS} and ARID1A^{PLD 30QtoG}) and d) BRG1^{PLD} (BRG1^{PLD}, BRG1^{PLD Aro+} and BRG1^{PLD Aro++}) as indicated. Hoechst was used to stain the cell nucleus, which is shown in blue. The phase separation capacity is quantified over various levels of nuclear protein concentrations. A phase separation (PS) score of ‘1’ indicates the presence of nuclear condensates and a PS score of ‘0’ represents diffused expression patterns. The shaded regions represent the transition concentrations. Asterisk ‘*’ denotes cytoplasmic concentration. (n = total of 28-55 cells from two replicates). Also see Fig. S4.

Associated text from the revised manuscript

Tyrosine residues play a dominant role in mSWI/SNF subunit PLD phase separation

The phase separation capacity of PLDs has been attributed to multivalent interactions predominantly mediated by the distributed aromatic and arginine residues^{18,28}. While SS18^{PLD} has a lower fraction of aromatic and arginine residues (0.11) than FUS^{PLD} (0.14; **Tables S1-3**), it possesses a greater number of aromatic and arginine residues (39) than FUS^{PLD} (24). To test if increasing the number of aromatic residues can improve the phase separation driving force of FUS^{PLD} without changing the overall sequence composition, we created a dimer of FUS^{PLD}, termed FUS^{2XPLD} (**Fig. 2a**), which possesses a total of 48 aromatic residues at a fixed fraction of 0.14. In contrast to the FUS^{PLD}, which remained diffused at all expression levels, we observed that FUS^{2XPLD} formed phase-separated condensates in the cell nucleus at a relatively low expression level (**Fig. 2b**) similar to the three mSWI/SNF subunit PLDs (**Fig. 1f**). The estimated intracellular saturation concentration of FUS^{2XPLD} was observed to be similar to that of ARID1A^{PLD}, ARID1B^{PLD}, and SS18^{PLD} (**Figs. 2b; S3a, S4**). Analogous to mSWI/SNF subunit PLD condensates, FRAP experiments revealed that FUS^{2XPLD} condensates have a high degree of dynamic behavior (**Fig. S3b**). To test whether the stronger driving force for phase separation of FUS^{2XPLD} primarily stems from the greater number of tyrosine residues and not simply from its increased length, we further created a variant of FUS^{2XPLD}, termed FUS^{2XPLD half YtoS}, where we replaced the tyrosine residues to serine in the second half of the FUS^{2XPLD}. This sequence variation led to a complete loss of phase separation FUS^{2XPLD} in living cells even at 10-fold higher intracellular concentrations (**Figs. 2a,b; S4**), implying that the number of tyrosine residues is a key determinant of phase separation in this PLD. Next, to test if tyrosine residues are also important for the phase separation of mSWI/SNF subunit PLDs, we first created an ARID1A^{PLD} variant where we mutated all 29 tyrosine residues to serine (29Y-to-S), termed ARID1A^{PLD YtoS}. We observed that 29Y-to-S substitution abolished the ARID1A^{PLD} phase separation in the cell at all expression levels (**Figs. 2c; S4**). Apart from tyrosine residues, ARID1A^{PLD} primary sequence shows enrichment of glutamines in the C-terminal region with multiple polyQ tracts with stretches of three to four Gln residues. Previous studies have suggested that polyQ regions can promote LCD self-association^{61,62}. However, when we mutated these Gln residues to Gly and created an ARID1A^{PLD} variant, termed ARID1A^{PLD 30QtoG}, we observed only a modest (~2 fold) increase in intracellular C_{sat} (**Figs. 2c; S4**). These data suggest that Tyr residues play a greater role in driving homotypic ARID1A^{PLD} phase separation, similar to the FUS^{PLD}, whereas polar residues such as Gln play a moderate role.

Based on the results obtained from the sequence perturbations of FUS^{2XPLD} and ARID1A^{PLD}, it appears that intracellular phase separation of PLDs can be tuned by the number of Tyr residues. Since BRG1^{PLD}, which only contains seven Tyr residues but a large number of proline residues, does not phase separate in cells, we attempted to improve its condensation driving force by increasing the tyrosine content. To this end, we created two variants where we mutated 17 proline and 41 proline residues to tyrosine residues, termed BRG1^{PLD Aro+} and

BRG1^{PLD Aro++}, respectively. We observed that both BRG1^{PLD} variants can form intracellular condensates with comparable C_{sat} to other mSWI/SNF PLDs. However, the BRG1^{PLD Aro++} was observed to form large irregular aggregates in the cytoplasm and was predominantly excluded from the nucleus (Figs. 2d; S4), which is likely due to strong homotypic interactions mediated by a large number of Tyr residues in this synthetic BRG1^{PLD} variant.

(iii) The cooperative co-condensation of two PLDs below C_{sat} is almost presented as a new result. There are many examples in the literature where this has been shown previously. An extreme example of such behavior is the complex coacervation between two oppositely charged proteins.

We respectfully disagree with the reviewer's assertion here. PLD mixtures are fundamentally different than complex coacervates since the complex coacervate forming systems phase separates solely via obligate heterotypic interactions. As we have shown in our study, mSWI/SNF PLDs have a strong propensity to undergo homotypic phase separation with saturation concentrations that depend on the primary sequence of respective PLD chains, as demonstrated for five naturally occurring PLDs that are important for the assembly and phase separation of transcriptionally relevant condensates. We show that despite such strong homotypic interactions, these functionally related PLDs co-phase separate and their heterotypic interactions are detectable in the dilute phase in the absence of phase separation. We have shown that rigorously in vitro with bead halo assay and in live cells with a light-inducible phase separation assay. These results are significant since they provide a direct measure of heterotypic LCD complexation in vitro and in live cells in a sequence-specific manner in the dilute phase.

We would also like to point out that understanding the rule of IDR-driven interactions between TFs and co-activators is of great importance as recently demonstrated by Ben Sabari and colleagues (DOI: <https://doi.org/10.1016/j.cell.2022.12.013>) and our results provide strong evidence how heterotypic PLDs can play a crucial role in this process. Moreover, following the suggestions of Reviewer 3, we have now included an additional non-prion-like IDR of a transcriptional repressor protein, FOXG1, which is similar in length as FUS-PLD, which did not positively partition in FUS-PLD and SS18-PLD condensates in vitro and in live cells. Interestingly, FOXG1-IDR does not have many Tyr residues, and the absence of interactions between this IDR and PLDs in the dilute phase (probed by bead halo assay) and dense phase (probed by partitioning to PLD condensates in vitro and in live cells) are consistent with the collective observations that Tyr residues are important for homotypic and heterotypic PLD interactions. These new results also support our model that the specificity of PLD-PLD interactions is important for scaffold-client interactions. We have added a new sub-section including Fig. 5 and Fig. S13-14 in the manuscript to describe these new results.

(iv) I find the paper making many conclusions or general statements that are not partially or fully supported by the preceding data. In other cases, these statements are not new or unexpected but presented as such. This makes it difficult to put the results of this paper in proper context.

There are no specific points to be addressed here. If there are concerns with valid reasonings, we are happy to address them as we have done with other referees' comments rigorously to the best of our abilities. Please refer to our response throughout this letter and the revisions made in response to the referees' comments. We think this study stands on its merit, as the other reviewers have found. Thank you.

(v) The authors may want to develop a simple theoretical model that takes into account the dilute

phase binding affinity between the same or different PLDs and how changing it will change the Csat. The paper only presents the expected behavior in a highly qualitative manner. The only way to make it interesting is to start the quantification of such effects that the condensate field is lacking in many respects.

We have added substantial amount of new data to strengthen the conclusions of the paper. The details are provided above. We think that quantitative model building should be best tackled in a future study.

(vi) I kept getting confused with the phrase "ternary PLD mixture" as many of the figures appear to show results from a binary mixture.

We thank the reviewer for raising this point. Ternary PLD mixtures are considered as three component mixtures: PLD-A, PLD-B, and the solvent. Since multiple reviewers have raised concern over the use of this term, we removed the use of ternary from the manuscript.

Reviewer #3 (Remarks to the Author):

The manuscript by Davis et al. "Heterotypic interactions in the dilute phase can drive co-condensation of prion-like low-complexity domains of FET proteins and mammalian SWI/SNF complex" is an interesting and well-written manuscript providing evidence that the PLD domains are important for interactions between chromatin remodeling components (SWI/SNF) and RNA-binding FET proteins. The latter are known to form recurrent fusion proteins in around 20 cancer types leading to aberrant transcription. This manuscript is therefore relevant for broad areas such as chromatin remodeling complexes and their interaction with transcription factors, transcription and when these processes are dysregulated in cancer. Davis et al. show that the "strength" of the PLD is sequence-dependent, mainly determined by the number of stickers (aromatic residues and arginine) which determine both phase separation capacity and co-partitioning. They note that heterotypic PLD interactions are very important, and in many cases stronger than homotypic interactions, which leads to complete mixing in the condensates and, as expected, lowering of saturation concentrations. For the first time, they also relate the PLD interactions observed in subsaturated concentrations in the dilute phase directly with co-condensation when condensation is induced. This is important since much debate is ongoing whether phase separation is required for many of biological processes such as transcriptional activation or if increased interactions of PLD proteins are enough. The manuscript is generally very well-written and the figures are clear. However, some concerns must be addressed. Specifically which parts of the PLD are really required for the interactions/phase separation? They show that the PLD is enough for interaction and condensate formation, but do relevant proteins such as SWI/SNF components without a PLD interact with and co-partition into condensates? Another option would be to infer mutations in the PLD such as exchanging tyrosines with serines and evaluate if interactions are lost.

We are very pleased to receive the positive comments of this Reviewer! They also raised several important issues that require clarification. In what follows, we provide detailed responses to the queries and comments. We have now performed additional experiments to provide evidence for the importance of aromatic residues as stickers in the PLD systems studied here by creating synthetic versions of various PLDs. We have also added a non-PLD IDR of a transcriptional repressor, FOXG1, and the folded domain of BRG1, the catalytic subunit of mSWI/SNF, in our experiments to demonstrate the specificity of PLD-PLD interactions. Below we describe these

new results and improved analysis/description of our results in the context of individual points raised by the reviewer.

Major remarks:

1. If you want to make strong conclusions regarding phase separation capacity of FET proteins and that PLDs are important for SWI/SNF assembly, some more experiments are needed (Figure 1C/F) to strengthen these claims, namely, to include EWSR1 and/or TAF15, and SMARCC1/2 in these analyses. SMARCC1/2 are very important for the assembly of the SWI/SNF core complex.

We appreciate the point raised by the reviewer here. To clarify, our work provided biophysical evidence that the disordered prion-like domains (PLDs) of multiple SWI/SNF complex subunits have a strong propensity to undergo phase separation in live cells and can directly mediate interactions with each other and with the FUS prion-like domain. In the context of FET fusion oncoproteins, such heterotypic interactions may drive the oncogenic transcriptional programs. The crux of our finding is that transcriptionally relevant interactions between a TF with a prion-like domain and the chromatin remodeler SWI/SNF complex can happen through sequence-specific dynamic multivalent interactions through disordered low-complexity PLDs. Based on our results, we also speculated that SWI/SNF prion-like domains may play a role in the assembly of the complex.

Following the reviewer's suggestions, we have performed two additional sets of experiments. First, we tested the intracellular phase behavior of TAF15 PLD and EWSR1 PLD in HEK293T cells and observed a stronger propensity of phase separation in both cases as compared to the FUS PLD, which remained diffused at all expression levels (**Fig. S4b**). **The results are provided below in point 2b.** This observation suggests distinct sequence grammars for phase separation of different FET PLDs. Second, we tested the phase separation capacity of SMARCC1^{PLD} in HEK293T cells and did not observe any phase separation at all expression levels, similar to FUS and BRG1 PLDs (**Fig. S4a**). This observation implies that SMARCC1^{PLD} has a very weak driving force to undergo phase separation. This is not surprising since our sequence analysis of SMARCC1^{PLD} shows that it has a very low number (2) of aromatic amino acids to drive phase separation. We note that, when our manuscript was under revision, a study reported that the deletion of IDRs of ARID1A/B, which included the prion-like domains, did not affect the assembly of the mSWI/SNF complex but suppressed the condensation and functional partner recruitment (Patil, Ajinkya, et al. Cell 186.22 (2023): 4936-4955. <https://doi.org/10.1016/j.cell.2023.08.032>). Based on this report and our study of multiple SWI/SNF PLDs, we speculate that cooperative interactions involving **multiple PLDs** may provide an emerging driving force for the assembly, phase separation, and function of the SWI/SNF complex. We have now included a short description highlighting this point in the discussion and cited this recently published paper.

“Moreover, our results on the mixtures of PLDs reducing the saturation concentration for phase separation of the individual components suggest that transcription factors and co-activators can enhance phase separation of each other through co-scaffolding (Fig. 8). This implies that multiple proteins with PLDs can provide a positive cooperative effect to reduce the concentration required for phase separation of the collection of proteins. We speculate that heterotypic PLD-mediated positive cooperativity in protein-protein interactions is likely to play key roles in the operation of SWI/SNF complexes and their interactions with transcription factors containing similar low-complexity domains”

2a. When you introduce how to study condensates in live cells, you describe that these PLDs could form condensates at low expression levels, and then say that you have done transient transfection. You need to be careful how to write about transient transfection and expression

levels. According to my experience, transient transfection does not lead to low expression values, and they rarely match endogenous levels. Please look over the text with this in mind and perhaps include in discussion.

We acknowledge the reviewer's concern that transient expression leads to a protein level that is often much higher than endogenous levels. The CMV promoter used in our constructs is a strong promoter in HEK293T cells (Qin, Jane Yuxia, et al., PloS one 5.5 (2010): e10611. <https://doi.org/10.1371/journal.pone.0010611>). When we use the term 'low expression', we are talking in relative terms, meaning it is usually in comparison to the various PLD constructs that are overexpressed in the same manner and not to the endogenous protein concentration. We have now revised the text to carefully point out that by low expression, we are speaking in relative terms. The revised text now reads

“Since mSWI/SNF PLDs show low micromolar C_{sat} values in vitro, we posited that they may form condensates in live cells at relatively low expression levels compared to FUS^{PLD} .”

2b. In figure 1F-G, I thought it was nice that you show that different expression levels (“concentrations”) of PLDs lead to either diffuse pattern or condensate formation. However, it would be interesting to see if condensates in the transition region forms smaller condensates than at larger concentrations. What concentration is the image from? For example, at endogenous expression levels, FUS-DDIT3 form many small condensates while overexpressed FUS-DDIT3-EGFP form few, large condensates (Owen et al., 2021). As pointed out, the level of PLDs is very important and overexpression changes their properties and perhaps function.

As requested by the reviewer, we have now included images at varying protein concentrations to denote changes in the number and size of PLD condensates with increasing protein levels (**Fig. S4**). As the relative protein concentration increases, we qualitatively observe an increase in condensate size & number for most PLDs. However, we refrained from performing more quantitative analysis since we observed that some PLDs, such as ARID1A/B^{PLD}, BRG1^{PLD Aro +}, and BRG1^{PLD Aro ++} form large irregular cytoplasmic inclusions at relatively higher expression levels. We clearly note that in the legend of **Fig. S4**.

a

Figure S4: Fluorescence microscopy images of HEK293T cells expressing GFP-tagged PLDs and variants of **a)** mSWI/SNF subunits, and **b)** FET proteins at varying expression levels. The mean GFP intensity values are noted in each image. Hoechst was used to stain the cell nucleus, which is shown in blue. For BRG1^{PLD Aro⁺⁺} and ARID1A^{PLD YtoS}, mean cytoplasmic intensities are noted since their expression was predominantly cytoplasmic. For some mSWI/SNF subunit PLDs, higher expression level has been noted to result in the formation of irregular cytoplasmic condensates.

2c. A minor remark, in Figure 1F-G. Is a random number of condensates included for the different PLDs? Please also include number of replicates (*n*) in figure legends for all experiments.

The figure legends have been updated accordingly.

3. An interesting result was how the length, or really the number of aromatic and arginine stickers, determines the phase separation capacity. This was described nicely in the text and discussion. However, the title of section 2 (The length of PLD chains determines their saturation concentration in live cells; line 189) is a bit misleading. Consider revising and substituting the word determines. Also, the same applies to line 456 in the discussion (strong phase separation driving forces arise from their longer chain lengths). Another option would be to investigate whether the same amount of aromatic and arginine stickers in PLDs of different lengths show similar capacity for condensate formation.

We thank the reviewer for raising this important point. We have revised our discussions on the chain length in the revised manuscript and replaced them with the results and discussions on the PLD sequence features that drive the phase separation and heterotypic co-condensation. Specifically, we have now included new data with new constructs to show that Tyr residues play a key role in the driving force for phase separation of the mSWI/SNF PLDs tested in the system. However, non-aromatic polar residues, such as glutamines also play a modest role. To this end, we have created five new constructs.

(1) A new variant of FUS^{2XPLD} where we substituted Tyr residues from the 2nd half of the FUS-PLD with Ser residues (FUS^{2XPLD halfYtoS}). Unlike the FUS^{2XPLD}, which phase separates in live cells at a relatively low expression level, FUS^{2XPLD halfYtoS} remained diffused at an order of magnitude higher expression level, similar to FUS^{PLD}. This is remarkable since FUS^{2XPLD halfYtoS} has twice the length of FUS-PLD and the only difference between this variant and FUS^{2XPLD} is that the former contains 24 Tyr residues and the latter contains 48 Tyr residues. This data suggests that the length alone is insufficient to drive phase separation of a PLD chain, it requires a critical number of aromatic residues in this case.

(2) We have made 29YtoS mutations in ARID1A^{PLD}, which abolished the phase separation capacity of AIRD1A^{PLD} in HEK293T cells, again suggesting the role of Tyr residues as stickers in driving phase separation in this system.

(3) We have made 30QtoG mutations in ARID1A^{PLD}, which increased the saturation concentration for phase separation of AIRD1A^{PLD} in HEK293T cells by ~ 2 fold, suggesting a modest role of non-aromatic residues in driving phase separation in this system.

(4) We have introduced 17 and 41 additional Tyr residues in WT-BRG1^{PLD} in two new synthetic variants, which we named BRG1^{PLD ARO+} and BRG1^{PLD ARO++}, respectively. Although WT-BRG1^{PLD} does not phase separate in the cell at any expression level, both BRG1^{PLD ARO+} and BRG1^{PLD ARO++} showed intracellular condensates, with BRG1^{PLD ARO++} showing predominantly large irregular clusters in the cytoplasm.

These new data have been added as **Fig. 2** and **Fig. S4** in the revised manuscript and the corresponding text and discussion have been modified/added.

Figure 2: Tyrosine residues play a dominant role in mSWI/SNF PLD phase separation. **a)** A bubble chart representation of the sequence composition of FUS^{PLD}, FUS^{2XPLD}, and FUS^{2XPLD half YtoS}. The lengths of the PLDs are displayed as amino acid count “aa”. Fluorescence microscopy images of HEK293T cells expressing GFP-tagged PLDs and variants of **b) FUS^{PLD}** (FUS^{PLD}, FUS^{2XPLD}, or FUS^{2XPLD half YtoS}), **c) ARID1A^{PLD}** (ARID1A^{PLD}, ARID1A^{PLD YtoS} and ARID1A^{PLD 30QtoG}) and **d) BRG1^{PLD}** (BRG1^{PLD}, BRG1^{PLD Aro+} and BRG1^{PLD Aro++}) as indicated. Hoechst was used to stain the cell nucleus, which is shown in blue. The phase separation capacity is quantified over various levels of nuclear protein concentrations. A phase separation (PS) score of ‘1’ indicates the presence of nuclear condensates and a PS score of ‘0’ represents diffused expression patterns. The shaded regions represent the transition concentrations. Asterisk ‘*’ denotes cytoplasmic concentration. (n = total of 28-55 cells from two replicates). Also see **Fig. S4**.

Associated text from the revised manuscript

Tyrosine residues play a dominant role in mSWI/SNF subunit PLD phase separation

The phase separation capacity of PLDs has been attributed to multivalent interactions predominantly mediated by the distributed aromatic and arginine residues^{18,28}. While SS18^{PLD} has a lower fraction of aromatic and arginine residues (0.11) than FUS^{PLD} (0.14; **Tables S1-3**), it

possesses a greater number of aromatic and arginine residues (39) than FUS^{PLD} (24). To test if increasing the number of aromatic residues can improve the phase separation driving force of FUS^{PLD} without changing the overall sequence composition, we created a dimer of FUS^{PLD}, termed FUS^{2XPLD} (**Fig. 2a**), which possesses a total of 48 aromatic residues at a fixed fraction of 0.14. In contrast to the FUS^{PLD}, which remained diffused at all expression levels, we observed that FUS^{2XPLD} formed phase-separated condensates in the cell nucleus at a relatively low expression level (**Fig. 2b**) similar to the three mSWI/SNF subunit PLDs (**Fig. 1f**). The estimated intracellular saturation concentration of FUS^{2XPLD} was observed to be similar to that of ARID1A^{PLD}, ARID1B^{PLD}, and SS18^{PLD} (**Figs. 2b; S3a, S4**). Analogous to mSWI/SNF subunit PLD condensates, FRAP experiments revealed that FUS^{2XPLD} condensates have a high degree of dynamic behavior (**Fig. S3b**). To test whether the stronger driving force for phase separation of FUS^{2XPLD} primarily stems from the greater number of tyrosine residues and not simply from its increased length, we further created a variant of FUS^{2XPLD}, termed FUS^{2XPLD halfYtoS}, where we replaced the tyrosine residues to serine in the second half of the FUS^{2XPLD}. This sequence variation led to a complete loss of phase separation FUS^{2XPLD} in living cells even at 10-fold higher intracellular concentrations (**Figs. 2a,b; S4**), implying that the number of tyrosine residues is a key determinant of phase separation in this PLD. Next, to test if tyrosine residues are also important for the phase separation of mSWI/SNF subunit PLDs, we first created an ARID1A^{PLD} variant where we mutated all 29 tyrosine residues to serine (29Y-to-S), termed ARID1A^{PLD YtoS}. We observed that 29Y-to-S substitution abolished the ARID1A^{PLD} phase separation in the cell at all expression levels (**Figs. 2c; S4**). Apart from tyrosine residues, ARID1A^{PLD} primary sequence shows enrichment of glutamines in the C-terminal region with multiple polyQ tracts with stretches of three to four Gln residues. Previous studies have suggested that polyQ regions can promote LCD self-association^{61,62}. However, when we mutated these Gln residues to Gly and created an ARID1A^{PLD} variant, termed ARID1A^{PLD 30QtoG}, we observed only a modest (~2 fold) increase in intracellular C_{sat} (**Figs. 2c; S4**). These data suggest that Tyr residues play a greater role in driving homotypic ARID1A^{PLD} phase separation, similar to the FUS^{PLD}, whereas polar residues such as Gln play a moderate role.

Based on the results obtained from the sequence perturbations of FUS^{2XPLD} and ARID1A^{PLD}, it appears that intracellular phase separation of PLDs can be tuned by the number of Tyr residues. Since BRG1^{PLD}, which only contains seven Tyr residues but a large number of proline residues, does not phase separate in cells, we attempted to improve its condensation driving force by increasing the tyrosine content. To this end, we created two variants where we mutated 17 proline and 41 proline residues to tyrosine residues, termed BRG1^{PLD Aro+} and BRG1^{PLD Aro++}, respectively. We observed that both BRG1^{PLD} variants can form intracellular condensates with comparable C_{sat} to other mSWI/SNF PLDs. However, the BRG1^{PLD Aro++} was observed to form large irregular aggregates in the cytoplasm and was predominantly excluded from the nucleus (**Figs. 2d; S4**), which is likely due to strong homotypic interactions mediated by a large number of Tyr residues in this synthetic BRG1^{PLD} variant.

4a. In the scaffold/client experiments in figure 3B, it would be nice to have microscopy images of the condensates before the client PLD is added. Also, it would be very valuable to have a negative control, preferably a SWI/SNF component without a PLD domain, to show that not “everything” is partitioning into the scaffold condensates, i.e. that partitioning is due to the PLD domain.

The images of condensates in **Fig. 1c** show the condensates without any client recruitment. The clients are added at a very low concentration, ~ 1 - 2% of the scaffold concentration, and did not change the morphology of the droplets.

Figure 1c

As the reviewer suggested, we have now included an additional non-prion-like IDR of a transcriptional repressor protein, FOXG1, which is similar in length to FUS-PLD, in our analysis. FOXG1^{N-IDR} did not partition in FUS-PLD and SS18-PLD condensates in vitro and in live cells (Fig. 4; Fig. S13). Finally, to test if PLDs alone can be sufficient to explain the enrichment of a full-length mSWI/SNF subunit in FUS^{PLD} condensates, we tested the folded domain of BRG1 for its ability to interact with FUS^{PLD} condensates. We observed that the GFP-tagged BRG1^{Folded} does not enrich within OptoFUS^{PLD} condensates while both BRG1^{PLD} and full-length BRG1 do (Fig. S14). These new results further support our claim on the specificity of PLD-PLD interactions. To describe the results of Figs. 4, S13, and S14, we have now added a new subsection in the revised manuscript “*PLD condensates exhibit specificity in interactions with functionally-linked IDRs*”

Figure 5: Selectivity in IDR interactions with condensates formed by prion-like domains. **a)** The PONDR score showing regions of disorder (>0.5) for the FOXG1 protein. The region shaded in red shows the N-terminus IDR for FOXG1 used in the study. **b)** Amino acid composition of FOXG1^{N-IDR}. The color codes for amino acids are provided in Table S6. **c)** Partitioning of AlexaFluor488 labeled FOXG1^{N-IDR} within condensates of FUS^{PLD} (250 μ M) and SS18^{PLD} (50 μ M), respectively. **d)** Enrichment is calculated as partition coefficient and displayed as a box-and-whisker plot for both FOXG1^{N-IDR} and BRG1^{PLD} within these condensates (n = total of 25-50 droplets from two replicates). **e)** HEK293T cells co-expressing SS18^{PLD} and mCherry-tagged FOXG1^{N-IDR} or GFP-tagged FOXG1^{N-IDR} and OptoFUS^{PLD} constructs. The degree of colocalization is displayed as intensity profiles for condensates shown in the inset images. Green represents the intensity profile of GFP-tagged protein and red represents the profile for mCherry-tagged protein.

Figure S13: a) Fluorescence microscopy image of HEK293T cells expressing GFP-tagged FOXG1^{N-IDR}. Yellow dashed lines indicate the nuclear periphery. **b)** Partitioning of AlexaFluor488 labeled FOXG1^{N-IDR} within condensates formed by ARID1A^{PLD} and ARID1B^{PLD} (50 μ M), respectively. **c)** Enrichment is calculated as partition coefficient and displayed as a box-and-whisker plot for both FOXG1^{N-IDR} and BRG1^{PLD} within these condensates (n = total of 25-50 droplets). **d)** HEK293T cells co-expressing GFP-ARID1A^{PLD} or GFP-ARID1B^{PLD} and mCherry-tagged FOXG1^{N-IDR}. The degree of colocalization is displayed as intensity profiles for condensates shown in the inset images. Green represents the intensity profile of the GFP-tagged construct and red represents the profile for mCherry-tagged construct.

Figure S14: HEK293T cells co-expressing BRG1^{PLD}, BRG1^{Folded} or Full-length BRG1 and mCherry-tagged OptoFUS^{PLD} construct. The degree of colocalization is displayed as intensity profiles for condensates shown in the inset images. Green represents the intensity profile of the GFP-tagged construct and red represents the profile for mCherry-tagged construct.

4b. The same is true in Figure 6B, D, E. It would strengthen your results to again include another negative control, such as SWI/SNF component without PLD for the bead halo assay to show that these are not enriched at the bead surface. For Figure 6D, E, a negative control is needed to show that only specific proteins co-localize directly when FUS condensates are induced.

This has been done. Please see our response to the previous comment with regard to a negative control for cellular assays. We have also included FOXG1^{N-IDR} for the bead halo assay (see below) as a negative control and updated Fig. 6b-c (now Fig. 7b-c in the revised manuscript) accordingly.

Figure 6: Heterotypic PLDs interact at sub-saturation concentrations. **a)** A schematic representation of the bead halo assay. Created with BioRender.com. **b)** 250 nM of AlexaFluor488-labeled His₆-MBP-constructs - SS18^{PLD}, BRG1^{PLD}, FOXG1^{N-IDR}, or control His₆-MBP was attached to Ni-NTA beads. 250 nM of FUS^{PLD} labeled with AlexaFluor594 was then added to the above beads (see Materials and Methods for further details). Binding was quantified using the ratio of fluorescence intensities (fluorescence signal from FUS^{PLD}/fluorescence signal from the scaffold) on the surface of the bead. **c)** A box-and-whisker chart of the intensity ratios is plotted with the mean and standard deviation (n = 8 beads/ trial).

5. In the discussion, the second to last section raises an interesting question (line 509) but there is a lack in the reasoning in the paragraph. On line 511, please make it clear if you refer to heterotypic interactions of the SWI/SNF complex. Then on line 513, what do you mean with “this idea”? I also do not understand why you introduce “new” experiments in the discussion and how the result of this experiment helps the conclusion further. Also, it has been known for at least 10 years that FUS has a diffuse pattern but co-localizes to FUS-DDIT3 condensates when the fusion is present (Owen et al., 2021; Thelin-Järnum, Göransson, Burguete, Olofsson, & Åman, 2002; Thomsen, Grundevik, Elias, Ståhlberg, & Åman, 2013). Instead, if you want to continue these discussion points and draw conclusions about FET fusion proteins, it would be much more worthwhile to perform more experiments with FUS-DDIT3-EGFP and other SWI/SNF PLDs, to determine their co-partitioning in condensates, and if the partition coefficient of FUS and FUS-DDIT3 differ. Although keep in mind that the FET proteins are present in live cells.

The reviewer’s point is well-taken here. We have now performed a colocalization analysis for the PLDs within FUS-DDIT3 condensates and the partition coefficients follow the same trend as PLD partitioning within SS18-PLD condensates. Since FUS does not form large condensates in cells due to the buffering effect of the nuclear RNA, we were not able to perform a similar analysis for colocalization. Moreover, we have added the suggested references (Reference # 86-88 in the revised manuscript) for showing colocalization of FUS and FUS-DDIT3 to replace the SI figure and changed the text to the following:

“The ability of the FET family of transcription factors containing an N-terminal PLD to orchestrate oncogenic gene expression has recently been linked to their aberrant interactions with the BAF complex subunits, such as BRG1^{40,45}. Previously, heterotypic interactions have been reported for transcription factors containing PLDs such as EBF1 and FUS³⁹, as well as FET proteins and FUS-DDIT3⁸³⁻⁸⁵. Given multiple mSWI/SNF subunits contain long PLDs, could such interactions be mediated by these intrinsically disordered LCDs (Fig. 8)? Indeed, our results suggest that mSWI/SNF subunit PLDs can engage in sequence-specific interactions with each other and with the PLD of FUS. We observe that FUS-DDIT3 condensates can enrich mSWI/SNF PLDs (Fig. S19) with an enrichment coefficient similar to what was observed for mSWI/SNF PLD-only condensates (Figs. 3d; S9). Since PLDs are common in many endogenous and cancer-associated fusion transcription factors, based on our results reported here, we speculate a common mode of functional protein-protein interactions in transcriptional regulation for these factors through sequence-specific heterotypic interactions between low-complexity protein domains.”

Figure S19: Enrichment coefficients of mCherry-tagged PLDs (ARID1A^{PLD}, ARID1B^{PLD}, BRG1^{PLD}, FUS^{PLD}, SS18^{PLD}, and mCherry alone) within condensates formed by GFP-FUS-DDIT3 in HEK293T cells. Enrichment is calculated as the ratio of mean intensities from the dense phase and the dilute phase. The average enrichment coefficient score is shown in red for each client. (n = 25-65 condensates from 4-8 cells). Student's t-test was used to calculate significance for each of the constructs and the p-values are tabulated.

6. One of the major concepts of this manuscript is the importance of the sequence-specificity versus the number of sticker amino acids. For example, the different co-partitioning of FUS^{PLD} and BRG1^{PLD}. Are they explained by other differences than the number of stickers? In addition, the bad phase competence of BRG1^{PLD}, is it only due to the lower amount of aromatic/arginine stickers? Furthermore, is it possible to separate sequence-specificity with phase separation capacity when it comes to PLD interactions? Please clarify and expand the discussion on these questions.

We thank the reviewer for raising this point. We have made new synthetic variants of the PLDs by mutating the aromatic stickers (Tyrosine residues) in BRG1^{PLD} and ARID1A^{PLD} to test the strengths of heterotypic interactions with FUS^{PLD}. Specifically, we have made the following variants:

(1) 29YtoS mutations in ARID1A^{PLD}, which abolished the partitioning of this PLD to optoFUS^{PLD} condensates in live cells.

(2) We have introduced 17 and 41 additional Tyr residues in WT-BRG1^{PLD} in two new synthetic variants, which we named BRG1^{PLD ARO+} and BRG1^{PLD ARO++}. Although WT-BRG1^{PLD} does not phase separate in the cell at any expression level, both BRG1^{PLD ARO+} and BRG1^{PLD ARO++} showed intracellular condensates. We observed increased colocalization (judged by the enrichment coefficient) of BRG1 variants into optoFUS^{PLD} condensates. Interestingly, BRG1^{PLD ARO++} showed predominantly large irregular clusters in cells, and some of them did not mix with FUS PLD condensates, likely due to the fact the these BRG1^{PLD ARO++} assemblies are predominantly solid-

like. These new data are included in the data in **Fig. 4** and provided below. Taken together with the data shown in Fig. 2, these results suggest that tyrosine residues are important for both homotypic and heterotypic interactions among mSWI/SNF PLD and FUS-PLD condensates.

Figure 4: Tyrosine residues are important for heterotypic interactions within mSWI/SNF PLD condensates. a) HEK293T cells co-expressing GFP-tagged ARID1A^{PLD}YtoS and mCherry-tagged OptoFUS^{PLD} construct. The degree of colocalization is displayed as intensity profiles for condensates shown in the inset images. Green represents the intensity profile of the GFP-tagged construct and red represents the profile for mCherry-tagged construct. **b)** HEK293T cells co-expressing PLDs (BRG1^{PLD}, BRG1^{PLD}Aro+, BRG1^{PLD}Aro++) and mCherry-tagged OptoFUS^{PLD} construct. The degree of colocalization is displayed as intensity profiles for condensates shown in the inset images. Green represents the intensity profile of the GFP-tagged construct and red represents the profile for mCherry-tagged construct. The

yellow line indicates the nuclear periphery. **c)** Enrichment coefficients of GFP-tagged BRG1 PLDs within mCherry-tagged OptoFUS^{PLD} condensates. Enrichment is calculated as the ratio of mean intensities from the dense phase and the dilute phase (n = 80-100 condensates from two replicates). Significance was calculated by a two-tail t-test; **** P ≤ 0.00001.

Associated text in the revised manuscript

Tyrosine residues are important for both homotypic and heterotypic interactions within mSWI/SNF PLD condensates

Results described in Figure 2 above suggest that Tyr residues play an important role in driving homotypic phase separation of mSWI/SNF PLDs as well as the FUS^{PLD}. Due to the strong propensity of these distinct PLDs to colocalize in condensates, we next aimed to investigate the role of tyrosine residues in heterotypic PLD co-condensation. We first examined heterotypic interactions between FUS^{PLD} and a variant of ARID1A^{PLD} lacking tyrosine residues. As discussed above, the native ARID1A^{PLD} condensates enrich FUS^{PLD} in live cells (**Fig. S10**). However, when OptoFUS^{PLD} was co-expressed with the 29Y-to-S variant of ARID1A (ARID1A^{PLD YtoS}), we observed that OptoFUS^{PLD} condensates did not enrich ARID1A^{PLD YtoS}, suggesting a loss of heterotypic interactions (**Fig. 4a**). To test if increasing the Tyr content of a PLD chain can increase the degree of co-localization, we performed a similar experiment with two BRG1^{PLD} variants containing 17 (BRG1^{PLD Aro+}) and 41 (BRG1^{PLD Aro++}) additional Tyr residues, respectively. We observe the BRG1^{PLD Aro+} variant has almost 2-fold greater enrichment within the OptoFUS^{PLD} condensates compared to the BRG1^{PLD}. For BRG1^{PLD Aro++} variant, however, we observed that the OptoFUS^{PLD} condensates that formed on blue light activation did not show enrichment of BRG1^{PLD Aro++}, but pre-existing condensates of BRG1^{PLD Aro++} were able to enrich OptoFUS^{PLD} to a greater extent than BRG1^{PLD} (**Fig. 4b-c**). Given BRG1^{PLD Aro++} variant formed large irregular cytoplasmic aggregates in the cell, this may imply that in this case, the BRG1^{PLD Aro++} homotypic interactions may have outweighed the heterotypic interactions, thereby driving the formation of discrete condensates similar to what was recently reported in the case of disordered FUS^{PLD} and LAF1^{RGG} systems⁴². It is also possible that BRG1^{PLD Aro++} condensates are viscoelastic solids, and their formation has resulted in a near-complete depletion of the soluble fraction of this PLD. Taken together with the data shown in Fig. 2, these results suggest that tyrosine residues are important for both homotypic and heterotypic interactions among mSWI/SNF PLD and FUS^{PLD} condensates.

Minor remarks:

7a. The text is generally clear and well-written, but paragraph 3 in the introduction can be improved such as the last sentence (line 81-83). Also on line 76, clarify what you mean with rich phase behavior.

We have rewritten this part of the text as follows for clarity:

“However, a key unanswered question is whether heterotypic PLD interactions occur at sub-saturation conditions, which may provide a mechanism for LCD-mediated functional protein recruitments such as interactions between FET oncofusions and mSWI/SNF complex in the absence of phase separation underlying oncogenic transcriptional programs.”

The phrase ‘rich’ has been replaced with ‘diverse’

7b. In the discussion, end of paragraph 3 and paragraph 4 should be rewritten for clarity. I think it would be good to merge these sections since they overlap in content.

We thank the reviewer for this comment. We have merged these two paragraphs and have revised them thoroughly to remove overlapping content. The revisions also include a discussion on the specificity within PLDs for partner interactions based on our finding that non-prion-like FOXG1^{N-IDR} do not enrich within PLD condensates in vitro and in live cells.

8. In Figure 1B, what does the color-coding mean? If it is based on amino acid properties, that would be nice, but this need to be clear from the figure or at least the figure legend.

Color coding for amino acids followed the categories below and was included in **Table S6** in the SI.

Side chain property	Amino acid	Color
Positively charged	Arg, Lys & His	Green
Negatively charged	Asp & Glu	Pink
Polar uncharged	Ser, Thr, Asn & Gln	Yellow
Hydrophobic	Ala, Val, Ile, leu, Met, Phe, Tyr & Trp	Blue
Others	Pro, Gly & Cys	Orange

9. Based on the FRAP in Figure 1E, is it really correct to write that all PLDs, including ARID1B, are rapidly exchanging molecules (line 140-141)?

To improve the clarity and remove any confusion, we have revised the text to the following:

“We next probed whether these condensates are dynamic using fluorescence recovery after photobleaching (FRAP) experiments. FRAP recovery traces indicate that all PLD condensates have liquid-like with varying diffusivity dynamics. (Fig. 1e). Based on the FRAP traces, we find that ARID1A^{PLD} forms the most dynamic condensates with more than 80% recovery, SS18^{PLD} is intermediate with ~60% recovery and ARID1B^{PLD} is the least dynamic with less than 40% recovery within the same observational timeframe.”

10. In result part 3 (line 223), the headline is misleading, you have not investigated any transcription factors in this section. FUS is not in itself a transcription factor, only in the oncogenic fusion form such as FUS-DDIT3.

We thank the reviewer for pointing this out. We have replaced the title of the section with the following:

“mSWI/SNF PLD condensates recruit low-complexity domains of the transcriptional machinery and RNA polymerase II via heterotypic PLD-mediated interactions”

11a. The enrichment coefficients from Figure 4A/S6 do not seem to match the images, for example the amount of SS18 that partitions into condensate for ARID1A and 1B looks different in the images. Are representative cells shown? Please indicate in all figure legends how many cells/condensates are used. Please clarify.

We thank the reviewer for this point. We would like to clarify that the cells and condensates shown in the enrichment profile are representative images and are part of the enrichment coefficient

analysis shown in **Fig. S9**. The enrichment coefficient quantification shows values of individual condensates as points (black dots). In the line profile image, this is just 1-2 representative condensates out of 36-85 condensates from 5-8 cells that we used for performing our statistical analysis. Indeed, we can see that the line profile data falls within the standard deviations of the coefficient quantification data. Moreover, the quantitative analysis was performed by using five random regions in the background for dilute phase intensity and this may not be clearly captured in the representative line profile image provided in **Fig. 3d** in the revised manuscript. Therefore, our line profile images are best served as a qualitative and representative depiction of heterotypic PLD co-localization in live cells whereas the enrichment coefficient quantification in **Fig. S9** provides a statistically meaningful picture.

Figure 3: Heterotypic PLDs interact and enrich within homotypic PLD condensates. d) HEK293T cells co-expressing GFP-SS18^{PLD} and either one of the mCherry-tagged PLDs (ARID1A^{PLD}, ARID1B^{PLD}, FUS^{PLD} and BRG1^{PLD}) or mCherry alone. The degree of colocalization is displayed as intensity profiles for condensates shown in the *inset* images. Green represents the intensity profile of GFP-SS18^{PLD} and red represents the profile for mCherry-tagged PLDs. The enrichment coefficients are reported in **Fig. S9**. The scale bar is 10 μm .

p -value	ARID1B ^{PLD}	BRG1 ^{PLD}	FUS ^{PLD}	mCherry
ARID1A ^{PLD}	7.49E-01	3.49E-15	1.56E-04	2.62E-29
ARID1B ^{PLD}		5.87E-05	2.48E-02	1.12E-08
BRG1 ^{PLD}			9.80E-05	5.58E-04
FUS ^{PLD}				2.00E-11

Figure S9: a) Enrichment coefficients of mCherry-tagged PLDs (ARID1A^{PLD}, ARID1B^{PLD}, BRG1^{PLD}, FUS^{PLD}, and mCherry alone) within condensates formed by GFP-SS18^{PLD} in HEK293T cells. Enrichment is calculated as the ratio of mean intensities from the dense phase and the dilute phase. The average enrichment coefficient score is shown in red for each PLD and the mCherry control. (n = 36-85 condensates from 5-8 cells). Student's t-test was used to calculate significance for each of the constructs and the p-values are tabulated.

11b. In Figure 4A/Figure S7/S8, I do not understand how you know that the scaffold protein labeled with GFP is the one forming the condensates since we know both PLDs cooperate with forming the condensates. Clarify this in the text.

We thank the reviewer for pointing this out and we agree that in cases where both the PLDs can form condensates in cells, it is not very clear to define the PLDs to be a scaffold. Therefore, in our analysis, we have performed co-localization analysis in both ways. To remove any confusion, we have revised the text as follows.

For in vitro experiments, we defined scaffolds and clients in each pair of PLD mixtures: scaffold is the protein that forms condensates ($C > C_{\text{sat}}$) and the client, defined as the protein that does not homotypically phase separate under the experimental conditions ($C < C_{\text{sat}}$), partitions into the scaffold condensates (**Fig. 3a; Fig. S7**).

For cellular experiments, in our first set of studies, we took advantage of the very low C_{sat} of SS18^{PLD}, which forms condensates robustly upon low expression in the cell nucleus and is considered the scaffold PLD for live cell experiments. Heterotypic PLD interactions within phase-separated GFP-tagged SS18^{PLD} condensates in live cells were probed by the degree of mCherry-tagged PLD co-partitioning. mCherry alone was used as a reference in our experiments.

We next performed similar experiments with ARID1A^{PLD} (**Fig. S10**) and ARID1B^{PLD} (**Fig. S11**) as scaffold condensates and made similar observations that except for BRG1^{PLD}, other PLDs strongly co-localize together in the dense phase.

11c. In some instances, in Figure 4A/Figure S7/S8, the scaffold protein (ARID1A, ARID1B) has more diffuse pattern when co-expressed with BRG1. Why? Does this point to interactions in the dilute phase? Or do BRG1 interfere with the scaffold protein in the condensates? Are these representative images? The same trend is seen in Figure S9 when co-expressed with Pol II. Here ARID1B do not seem to form condensates in the nuclei. These questions should be addressed.

We see that in cases of co-expression with mCherry-tagged ARID1A^{PLD} and ARID1B^{PLD} with GFP-tagged SS18^{PLD}, there is a slight reduction of ARID1A/B^{PLD} condensates. The same is true for the co-expression of other PLD pairs, where we qualitatively see a modest overall reduction of condensates in cells, which may be due to dilute phase interactions as proposed by the reviewer. To explain these observations qualitatively, we hypothesize that in certain cases, especially in the cellular microenvironment, heterotypic PLD interactions in the dilute phase can compete with the same in the condensed phase. This thermodynamic phenomenon has recently been discussed using the framework of polyphasic linkage (Ruff et al. 2021, <https://doi.org/10.1073/pnas.2017184118>). Further in-depth studies are needed to explore this systematically, which is best reserved for a follow-up study. However, despite this effect, we still see a strong enrichment of mCherry-tagged ARID1A/B^{PLD} in GFP-tagged SS18^{PLD} condensates (Fig. S9). Further, since we have supplemented our live cell results with recombinantly purified proteins (Fig. 3b; Fig. S7), we believe it does not impact our interpretation of the data. Future studies can also evaluate the effect of fluorescent protein tags and small molecule fluorophores on this phenomenon.

12. Please explain what you mean with a ternary PLD mixture, the first time mentioned in introduction and/or results. Also clarify what you mean with the following sentence or rephrase, “co-mixing of individual PLD phases within ternary PLD condensates” (line 346-347).
13. In the discussion, you describe that FUS and SWI/SNF PLD have positive cooperativity. Do you know of any examples of PLD-PLD interactions that do not have positive cooperativity (line 488) and what does that lead to? Consider discussing.

We thank the reviewer for raising this point. Ternary PLD mixtures are considered 3 component mixtures: PLD-A, PLD-B, and the solvent. Since multiple reviewers have raised concerns over the use of this term, we believe it would be best to remove it from the manuscript.

We have also modified lines 346-347 with the following sentence for improved clarity:

“The mixing of individual PLD phases and the formation of PLD co-condensates (Fig. 6d) was not only observed in vitro but also in live cells when two PLDs were co-expressed (Figs. 3d & S10-11)”

In the revised manuscript, we now show that when we co-express optoFUS^{PLD} and BRG1^{PLD Aro++} variant (a variant of BRG1^{PLD} that contains 41 additional Tyr residues; Table S4), they form co-existing condensates in cells instead of co-condensates. We interpret this data based on the relative strength of homotypic and heterotypic interactions present in a mixture of two PLDs. In this case, the homotypic interactions between BRG1^{PLD Aro++} molecules are much stronger than heterotypic interactions between BRG1^{PLD Aro++} and optoFUS^{PLD}, leading to discrete coexisting phases. We have now added the following text when we discuss these results:

In the Introduction section:

“In general, in multi-component mixtures of multivalent LCDs, homotypic and heterotypic interactions between LCD chains can either positively cooperate, negatively cooperate, or form coexisting phases, resulting in a diverse phase behavior and dense phase co-partitioning. The interplay between the specificity

and strengths of homotypic and heterotypic interactions dictates the co-condensation versus discrete condensate formation in an LCD sequence-specific manner⁴².”

In the Results section:

Given $BRGI^{PLD Aro^{++}}$ variant formed large irregular cytoplasmic aggregates in the cell, this may imply that in this case, the $BRGI^{PLD Aro^{++}}$ homotypic interactions may have outweighed the heterotypic interactions, thereby driving the formation of discrete condensates similar to what was recently reported in the case of disordered FUS^{PLD} and $LAF1^{RGG}$ systems⁴².

In the Discussion section:

If the homotypic interactions are substantially different in a mixture of IDRs, they can form coexisting dense phases with differential densities^{42,52,68,69}. This is likely the case for the $BRGI^{PLD Aro^{++}}$ variant, which formed discrete condensed phases when co-expressed with $OptoFUS^{PLD}$ condensates (**Fig. 4b-c**). However, except for this synthetic variant, our co-phase diagrams in vitro (**Figs. 6; S15**) and PLD co-condensation results in live cells suggest that heterotypic interactions between $mSWI/SNF$ PLD chains are dominant over the homotypic interactions in PLD mixtures.

References:

- Owen, I., Yee, D., Wyne, H., Perdikari, T. M., Johnson, V., Smyth, J., . . . Shewmaker, F. (2021). The oncogenic transcription factor FUS-CHOP can undergo nuclear liquid-liquid phase separation. *Journal of Cell Science*, 134(17). doi:10.1242/jcs.258578
- Thelin-Järnum, S., Göransson, M., Burguete, A. S., Olofsson, A., & Åman, P. (2002). The myxoid liposarcoma specific TLS-chop fusion protein localizes to nuclear structures distinct from PML nuclear bodies. *International Journal of Cancer*, 97(4), 446-450. doi:10.1002/ijc.1632
- Thomsen, C., Grundevik, P., Elias, P., Ståhlberg, A., & Åman, P. (2013). A conserved N-terminal motif is required for complex formation between FUS, EWSR1, TAF15 and their oncogenic fusion proteins. *FASEB Journal*, 27(12), 4965-4974.

These references have been cited.

REVIEWERS' COMMENTS

Reviewer #1 (Remarks to the Author):

The authors of Davis et al. have addressed my concerns. The added replicates and controls to the bead-halo assay provides much stronger support to their conclusions on interaction strength between FUSPLD and the tested proteins.

Reviewer #3 (Remarks to the Author):

Davis et al. have done a comprehensive revision with new experiments after the reviewers' suggestions that answers the majority of questions raised by the reviewers and strengthen the conclusions of the article. Specifically, they show that heterotypic interactions drive PLD co-condensation in live cells, the importance of tyrosine residues for homo- and heterotypic interactions, and the sequence specificity of condensation including the lack of interaction for a non-PLD transcriptional repressor. Therefore, I believe the article is suitable for publication. Some minor text edits/changes are encouraged.

- Optimize the figure space between panels, in some figures it is hard to see what belong to what figure.
- SMARCA4 is used in the figure when BRG1 is used throughout the manuscript.
- Results with optoFUS is now used before it is introduced.
- Figure S4 is referenced earlier than figure S3.
- Explain in figure legend where BRG1 PLD data comes from in Fig5/S13.
- In figure S16, wrong figure reference.
- The new colocalization studies of SWI/SNF PLD with FUS-DDIT3 condensates were interesting. Consider including representative images of the co-condensates.
- Importantly, in the discussion (p644-648), the references do not match the text that well. For example, where is the reference for EBF1? References 84-85, although relevant, do not describe the aberrant role of FET oncoproteins with SWI/SNF. It is also a bit confusing which heterotypic interactions you refer to, consider revising for better clarity.

We thank all the reviewers for their feedback. Below we provide a point-by-point response to the comments provided by Reviewer # 3 (our responses are in blue). The text reproduced from the manuscript is shown in red.

Reviewer #3 (Remarks to the Author):

Davis et al. have done a comprehensive revision with new experiments after the reviewers' suggestions that answers the majority of questions raised by the reviewers and strengthen the conclusions of the article. Specifically, they show that heterotypic interactions drive PLD co-condensation in live cells, the importance of tyrosine residues for homo- and heterotypic interactions, and the sequence specificity of condensation including the lack of interaction for a non-PLD transcriptional repressor. Therefore, I believe the article is suitable for publication. Some minor text edits/changes are encouraged.

Author's response: We are very pleased to receive the positive comments of this Reviewer! Below we have provided a point-by-point response to the minor concerns of the reviewer.

•*Optimize the figure space between panels, in some figures it is hard to see what belong to what figure.*

Author's response: We have optimized the figure size and organization of panels in each figure to improve the clarity and presentation of our results to the best of our ability.

•*SMARCA4 is used in the figure when BRG1 is used throughout the manuscript.*

Author's response: We have replaced SMARCA2/4 with "BRM/BRG1" in **Fig. 1a** as the reviewer suggested to match the naming of these subunits in the manuscript.

•*Results with optoFUS is now used before it is introduced.*

Author's response: We have now introduced optoFUS in the first instance of its occurrence.

•*Figure S4 is referenced earlier than figure S3.*

Author's response: The order of these two figures has been changed in the SI to resolve this issue.

•*Explain in figure legend where BRG1 PLD data comes from in Fig5/S13.*

Author's response: We have added the information indicating the source of the data for BRG1^{PLD} in the figure legends for **Fig. 5** as "(data reported in **Fig. 3b-c**)" and for **Fig. S13** as "(data reported in **Fig. S7**)"

•*In figure S16, wrong figure reference.*

Author's response: We thank the reviewer for pointing out this error. The figure reference has now been corrected.

•*The new colocalization studies of SWI/SNF PLD with FUS-DDIT3 condensates were interesting. Consider including representative images of the co-condensates.*

Author's response: The representative images have now been included in **Fig. S19a**. This is reproduced below:

Figure S19: a) HEK293T cells co-expressing GFP-FUS-DDIT3 and either one of the mCherry-tagged PLDs (ARID1A^{PLD}, ARID1B^{PLD}, BRG1^{PLD}, FUS^{PLD}, and SS18^{PLD}) or mCherry alone. The degree of colocalization is displayed as intensity profiles for condensates shown in the inset images. Green represents the intensity profile of GFP-FUS-DDIT3 and red represents the intensity profile for mCherry-PLD (or mCherry alone; bottom panel). The nuclear periphery is highlighted in yellow.

•Importantly, in the discussion (p644-648), the references do not match the text that well. For example, where is the reference for EBF1? References 84-85, although relevant, do not describe the aberrant role of FET oncoproteins with SWI/SNF. It is also a bit confusing which heterotypic interactions you refer to, consider revising for better clarity.

Author's response: The reference for EBF1 is ref # 40. By heterotypic in this case, we are referring to the full-length proteins that contain the PLDs. For better clarity, we have revised this statement, which is reproduced below:

“Previously, heterotypic protein-protein interactions have been reported for transcription factors containing PLDs such as EBF1 and FUS⁴⁰, as well as FET proteins and FUS-DDIT3^{83–85}.”